



# Estimating maximum mineral associated organic carbon in UK grasslands

Kirsty C. Paterson[1,2], Joanna M. Cloy[1], Robert. M. Rees[1], Elizabeth M. Baggs[2], Hugh Martineau[3], Dario Fornara[4], Andrew J. Macdonald[5], and Sarah Buckingham[1]

[1] Scotland's Rural College, West Mains Road, Edinburgh, EH9 3JG, United Kingdom.
[2] Global Academy of Agriculture and Food Security, Royal (Dick) School of Veterinary Studies, University of Edinburgh, Easter Bush Campus, Midlothian, EH25 9RG, United Kingdom.
[3] Treberfydd Farm, Wales, United Kingdom.
[4] Agri-Food & Biosciences Institute (AFBI), Newforge Lane, BT9 5PX, Belfast, United Kingdom.
[5] Sustainable Agriculture Sciences Department, Rothamsted Research, Harpenden, Hertfordshire, AL5 2JQ, United Kingdom.

*Correspondence to*: Sarah Buckingham (sarah.buckingham@sruc.ac.uk)

**Abstract.** Soil organic carbon (SOC) sequestration across agroecosystems worldwide can contribute to mitigate the effects of climate change by reducing levels of atmospheric $CO_2$. Mineral associated organic carbon (MAOC) is considered an important long-term store of SOC and the saturation deficit (difference between measured MAOC and estimated maximum MAOC) is frequently used to assess SOC sequestration potential following the linear regression equation developed by Hassink, (1997). However, this approach is often taken without any assessment of the fit of the equation to the soils being studied. The statistical 20 limitations of linear regression have previously been noted, giving rise to the proposed use of boundary line (BL) analysis and quantile regression (QR) to provide more robust estimates of maximum SOC stabilisation. The objectives of this work were to assess the suitability of the Hassink, (1997) equation to estimate maximum MAOC in UK grassland soils of varying sward ages and to evaluate the linear regression, BL and QR methods to estimate maximum MAOC. A chronosequence of 10 grasslands was sampled, in order to assess the relationship between sward age (time since last reseeding event) and current 25 and predicted maximum MAOC. Significantly different regression equations show that the Hassink, (1997) equation does not accurately reflect maximum MAOC in UK grasslands when determined using the proportion of fine soil fraction and current MAOC. The QR estimate of maximum SOC stabilisation was almost double that of linear regression and BL analysis (0.89 ± 0.074, 0.43 ± 0.017 and 0.57 ± 0.052 g C kg⁻¹ soil, respectively). Sward age had an inconsistent effect on the measured variables and potential maximum MAOC. MAOC across the grasslands made up 4.5 to 55.9% of total SOC, implying that there may be 30 either high potential for additional C sequestration in the mineral fraction of these soils, or stabilisation in aggregates is predominant in these grassland soils. This work highlights the need to ensure that methods used to predict maximum MAOC reflect the soil *in situ*, resulting in more accurate assessments of carbon sequestration potential.





## 1. Introduction

Carbon (C) sequestration in soils offers a significant opportunity to remove $CO_2$ from the atmosphere and store it into long lived C pools (Lal, 2004; Powlson et al., 2011), with co-benefits for soil structure and functioning (Lorenz and Lal, 2018; Smith, 2012; Soussana et al., 2004). Carbon sequestration refers to the removal of $CO_2$ from the atmosphere into long lived soil C pools, which would not otherwise occur under current management practices (Lal, 2004; Powlson et al., 2011). Soil organic carbon (SOC) is stabilised by three mechanisms i) inherent chemical recalcitrance, ii) adsorption to mineral surfaces,

and iii) occlusion of SOC within soil aggregates. With respect to SOC sequestration the mineral associated organic carbon (MAOC) in the fine soil fraction (< 20 μm) is often regarded as the most important due to its longer residence time (Baldock and Skjemstad, 2000; Six et al., 2002).

Human-managed grasslands are the dominant land use in the UK, covering 36% of the land area (Ward et al., 2016). Managed grasslands are planted and maintained to increase agricultural productivity through fertiliser and liming applications,

and the re-seeding of swards. They are thought to have high potential for sequestering more C  (Smith, 2014), however frequent re-seeding may result in changes in soil structure, nutrient cycling and SOC mineralisation (Carolan and Fornara, 2016; Drewer et al., 2017; Soussana et al., 2004). The long-term effect of re-seeding on SOC is understudied but is likely to affect physical soil aggregates making MAOC accessible for microbial mineralisation, and enhance the potential for SOC losses. It is therefore important to understand how disturbance might affect MAOC and thus the SOC sequestration ability of managed grasslands.

To utilise soils as a $CO_2$ drawdown mechanism, accurate estimates of their storage capability are required. It is well accepted that there is an upper protective capacity limit, or saturation point of MAOC (Six et al., 2002; Stewart et al., 2007). The ability to predict this saturation point is essential in order to assess the feasibility of SOC sequestration targets. Hassink, (1997) compared pairs of Dutch arable and grassland soils and found that while soil bulk SOC contents significantly differed among soils, MAOC did not. A positive relationship between the mass proportion of the fine soil fraction and associated C

and N concentrations in temperate and tropical soils was also observed. These findings led to the idea that the saturation point of the fine soil fraction could be estimated by linear regression using the mass proportion of fine fraction in a soil sample (%) and the current MAOC (g kg$^{-1}$ soil). With this approach, potential SOC sequestration (or saturation deficit) can be estimated by subtracting the current MAOC from the estimated maximum MAOC (MAOC$_{max}$) (Angers et al., 2011).

Several iterations of the concept have been proposed to overcome the limitations of linear regression. For example,

boundary line analysis (BL) uses a defined upper or lower subset of a data set to estimate the boundary line, when a limiting response to an independent variable(s) along a boundary is supported (Lark and Milne, 2016; Schmidt et al., 2000). Using the upper 90[th] percentile of a data set, BL analysis overcomes the limitation of linear regression depicting the mean response to

**Biogeosciences** Open Access
Discussions
**EGU**

the independent variable (Feng et al., 2013; Shatar and Mcbratney, 2004), which is thought to cause an underestimation of

sequestration potential. Quantile regression (QR) estimates the response of a specific quartile using the entire data set. It also

makes no assumptions regarding homogeneity of variance, thus increasing the robustness of the estimated $MAOC_{max}$, as sample

size is not reduced as in BL analysis (Beare et al., 2014; Cade and Noon, 2003). Using a forced zero intercept overcomes the

contradiction of a positive intercept indicating the presence of MAOC without any fine soil fraction (Beare et al., 2014; Feng

et al., 2013; Liang et al., 2009). These suggestions have been proposed to improve estimates of $MAOC_{max}$. However, several

studies use the original equation presented by Hassink, (1997) to estimate sequestration potential at different scales (e.g. Angers

et al., 2011; Chen et al., 2019; Lilly and Baggaley, 2013; Wiesmeier et al., 2014). This is frequently done without any validation

checks to determine the suitability of the Hassink, (1997) linear regression equation to predict $MAOC_{max}$ in the respective

studies.

The objectives of this study were (i) to assess the suitability of the Hassink, (1997) equation to estimate $MAOC_{max}$ in UK

grassland soils of varying sward ages, (ii) to evaluate the linear regression, BL and QR methods to estimate $MAOC_{max}$, and

(iii) to explore the relationship between sward age and current and predicted maximum MAOC. We hypothesised that i) the

linear regression equation developed using UK grassland soils would be significantly different to that of Hassink, (1997), and

that ii) grasslands with an older sward age, would have a greater proportion of total SOC stabilised as MAOC and a lower

sequestration potential.

**2. Materials and Methods**

**2.1 Site Description and Sampling**

Ten grassland chronosequences covering a wide range of soil types, land use and climatic conditions were identified across

the UK in 2016. The sites included the range of agricultural activity associated with UK grasslands (upland grazing, dairy, and

mixed grazing), variations in soil type (organo-mineral, mineral and chalk) and the majority of UK climatic zones (Table 1).

At each location, five to eight individual fields of different sward age (represented by years since a ploughing and reseeding

event), ranging from 1 to 179 years, were identified for sampling. In each field, areas were avoided which had different

applications of manure, soil types or topography, headlands, areas near gates, where lime or manure had previously been

dumped, or where livestock congregate.  Two replicate soil cores were collected to a depth of 30 cm using a soil auger with a

2.5 cm diameter steel core and bulked to give a single composite sample. This was repeated 10 times in each field at regular

intervals in a 'W' shape across the field totalling 10 replicate samples per field per site. Intact soil cores for determining bulk

densities were collected at three locations in each field at two depths (10 to 15 cm and 20 to 25 cm) using intact rings (7.5 cm

diameter, 5 cm height). Replicate samples were sieved to 2 mm and fresh subsamples were used to determine soil pH in water.



Remaining sieved soils were dried at 40°C and ball milled prior to determination of total C and N contents (% by mass) using a Flash 2000 elemental analyser. Intact soils were dried at 107°C and weighed to calculate dry bulk densities, any stones were

removed.

## 2.2 Soil fractionation

The fine fraction (< 20 μm) of the soil was separated using an adapted method of Hassink, (1997). Briefly, 20 g of dried sieved soil was soaked in 100 mL of deionised water for 24 hours. The suspension was then sonicated with a Microson XL2000

Ultrasonicator for 20 minutes at 20 W in 50 mL centrifuge tubes, surrounded by ice to prevent overheating. The separated samples were recombined in 150 mL tubes, and shaken end over end to disperse the soil water suspension. Sedimentation times were determined using a table applying Stokes Law, for 20 μm particles, a particle density of 2.65 g cm$^{-3}$ and sedimentation depth of 5 cm at temperatures between 20°C and 35°C (Jackson, 2005). After the appropriate sedimentation time, the fine fraction was siphoned off the soil suspension. The fine fraction was dried for 24 hours at 107°C and ball milled

prior to total C and N analysis (% by mass) using a Flash 2000 Elemental Analyser, to determine the current MAOC. At each site, a minimum of 3 fields varying in age (young, intermediate, and old at that location) were selected, and 3 of the 10 replicate field samples were selected at random for fractionation.

Hydrochloric acid (HCl) fumigation was used to remove carbonates from the Plumpton samples. Ball-milled samples, in silver capsules, were moistened with deionised water (1:4 sample:water ratio) to aid the efficiency of carbonate removal by

HCl fumes (Dhillon et al., 2015). The samples were placed in a vacuum desiccator with a beaker of 100 mL of 12 M HCl, for 24 hours and subsequently dried in a ventilated oven at 60°C for 16 hours, to remove excess moisture and HCl (Dhillon et al., 2015). Total C and N contents were determined as outlined above.

## 2.3 Statistical analyses

All statistical analyses were carried out using R software version 3.5.3 (Team, 2019). Significant differences were determined by ANOVA's and by post-hoc Tukey tests ($\alpha$ = 0.05). Where assumptions of normality and variance were not satisfied by testing (Shapiro Wilkens and Levenes Test) significant differences were identified using Kruskal test and post hoc Dunn test. A Kendal tau ($\tau$) correlation matrix was produced using the 'corrplot' package (Wei and Simko, 2017).



### 2.3.1 Regression analyses

Linear regression was used to predict $MAOC_{max}$ with the mass proportion of fine fraction (< 20 μm, %) in a sample and the measured MAOC (g C $kg^{-1}$ soil) as the independent and dependent variables, respectively. Regression equations were developed for the combined UK data set, and the individual sites. Linear regression with a forced zero intercept was used with data from this study and the data published in Hassink (1997).

BL analyses were performed as an alternative to linear regression, both with and without a forced zero intercept to predict $MAOC_{max}$ for all UK sites. The data was organised by mass proportion of the fine fraction (%) and divided into subgroups at 5, 10 and 15% intervals. The 10% interval reflects the method of Feng et al. (2013), whilst the 5 and 15% intervals were used to assess the effect of interval on estimation of $MAOC_{max}$. The groups were then ordered by measured MAOC (g C $kg^{-1}$ soil), and the values in the 90th percentile were used to plot the boundary line. BL analysis was not used for individual sites as it resulted in too few data points. QR analysis was performed in Rstudio using the 'quantreg' package (Koenker, 2019), for the 90th and median percentiles ($\tau = 0.90$ and $\tau = 0.50$). Significant differences between slopes were identified using the 'lsmeans' package (Lenth, 2016), followed by post-hoc Tukey tests ($\alpha = 0.05$).

### 2.3.2 Carbon saturation ratio

The carbon saturation ratio was determined in order to identify the degree of saturation across the sites, when estimating $MAOC_{max}$ using the Hassink (1997), UK, and site-specific linear regression equations both with and without a forced zero intercept, and the equations generated by BL and QR analyses. The carbon saturation ratio was calculated by dividing the current MAOC by the estimated $MAOC_{max}$. Values < 1 were deemed under saturated, = 1 as at saturation and > 1 as oversaturated.

## 3. Results

### 3.1 Current C concentrations

The measured total SOC and MAOC concentrations exhibited variation within the grassland sites (Fig. 1). Total SOC varied from 8.2 to 85.84g C $kg^{-1}$ soil, with a median of 32.72 g C $kg^{-1}$ soil. Hillsborough, Overton and Plumpton had significantly higher total SOC, whilst Harpenden and Llangorse had the lowest total SOC ($P < 0.05$) (Fig. 1). The measured MAOC ranged from 1.37 to 20.89 g C $kg^{-1}$ soil, with a median of 6.21 g C $kg^{-1}$ soil. Overton had the highest total MAOC ($P < 0.05$) and was the only organically managed site (Fig. 1). The proportion of OC stored as MAOC had high variability across the UK sites accounting for 4.5 to 50.12% of total SOC with a median of 17.51%. The proportion of total SOC stored as MAOC, and





proportion of fine fraction in a sample did not significantly differ in Harpenden and Overton, however they have significantly
different current MAOC (g C kg⁻¹ soil) ($P < 0.05$), indicating different saturation potentials (Fig. 1). Full details of all the
measured properties of bulk and fine fraction, per field are presented in Table A1.

The significance of correlations between the measured soil properties, time since reseeding and known environmental
factors were analysed. The matrix of Kendall tau ($\tau$) correlation coefficients in Table 2, revealed that current MAOC was
positively correlated with median annual temperature ($\tau = 0.13$, $P < 0.05$), %N ($\tau = 0.26$, $P < 0.0001$) and %C ($\tau = 0.27$, $P <$
$0.0001$) in the bulk soil, and negatively correlated with mean annual rainfall ($\tau = -0.36$, $P < 0.0001$), and %N ($\tau = -0.15$, $P <$
$0.05$) in the fine fraction.

Mass proportion of fine fraction and measured MAOC were positively correlated in cambisols ($R^2 = 0.61$, $P < 0.05$),
gleysols ($R^2 = 0.76$, $P < 0.05$), podzols ($R^2 = 0.93$, $P < 0.05$), and stagnosols ($R^2 = 0.88$, $P < 0.05$) (Fig. 2). However, the
proportion of MAOC to $SOC_{total}$ was greatest in luvisols ($P < 0.05$) (Fig. 3).


## 3.1 Estimated maximum MAOC

The slope generated from the UK data used to estimate MAOC (Table 3) was significantly different ($P < 0.05$) to the slope
reported in Hassink (1997). There was no significant difference between the slopes generated from the UK data, the data from
Hassink (1997) when estimated by linear regression with a forced zero intercept. Significantly different ($P < 0.05$) slopes were
found between the individual UK sites, owing to the range in the proportion of the fine fraction within each sample, from 1.85
to 51.8%, (Tables A3 and A4).

Coefficients from BL analysis are presented in (Table 3). There was no significant difference in slopes between the
5, 10, and 15% fine fraction intervals used. The median percentile QR analysis had a similar slope to the BL and linear
regression with forced zero intercept. QR using the 90th percentile resulted in the steepest slope of all estimation methods
(Table 3). The C saturation ratios revealed the difference in number of samples with potential to sequester more C (Table 4).
The Hassink (1997) linear regression equation, without a forced zero intercept, predicted the greatest number of unsaturated
sites, followed by the 90th percentile QR, with a forced zero intercept. There was no clear relationship between oversaturated
sites and proportion of silt and clay contents as oversaturation occurred across all proportions, indicated by points above the
lines in Fig. 4.


## 3.2 Effect of sward age on current C concentrations and estimated maximum MAOC

Sward age had a weak positive correlation with the mass proportion of the fine fraction (%) (Table 2). When grouped in five
year intervals, significant differences were found between age group and the mass proportion of the fine fraction (%), current





MAOC (g C kg$^{-1}$ soil), and the C:N ratio of the fine fraction (Table 5), however there was no consistent increase or decrease
with sward age. At the individual sites, significant differences were observed between fields, with some properties, but again
there was no consistent effect of sward age (Tables A3 and A4).

## 4. Discussion

### 4.1 Estimation of maximum mineral associated organic carbon

Determining the potential C sequestration capacity of soils is essential to predict the influence of land management for climate
change mitigation. The determination of saturation deficit using the mass proportion of the fine fraction and current MAOC is
an established method with a strong grounding in correlation between the variables. However, despite the wide use of the
Hassink (1997) linear regression equation, it has undergone very little further testing to determine suitability in other soils.
The significantly different slopes for the linear regression equations (Table 3) shows that the Hassink (1997) regression
equation is not suitable for estimating MAOC$_{max}$ in UK grasslands. Previous concerns have focused on the potential for the
equation developed by Hassink (1997) to underestimate MAOC$_{max}$, as linear regression represents the mean response of the
independent variable, rather than the maximum. For the UK grasslands in this study estimating MAOC$_{max}$ using the Hassink
(1997) regression approach resulted in a significant overestimation of MAOC$_{max}$ sequestration potential. Future work using
MAOC$_{max}$ predictions equations reported in the literature ( e.g. Beare et al., 2014; Feng et al., 2013; Hassink, 1997; Six et al.,
2002) should first conduct a validity test to ensure results are not significantly over or underestimated. To overcome the
contradiction of an intercept greater than zero, indicating that C is stabilised as MAOC without any fine fraction, a forced zero
intercept was used. The linear regression slopes with a forced zero intercept were not significantly different, and were similar
to that of Feng et al. (2013), 0.42 ± 0.002.. Liang et al. (2009) reported a lower slope of 0.36 in Chinese black soils, whilst
Beare et al. (2014) reported a slope of 0.70 ± 0.03 in long-term New Zealand pastures. The range of reported values, and
differences across the UK sites (Tables A3 and A4), suggest that the effect of the proportion of fine fraction of a sample on
MAOC is not consistent and likely reflects differences in pedogenic and environmental conditions, and land management.

Boundary line (BL) analysis and quantile regression (QR) have been suggested as alternatives to overcome the
limitations of linear regression. The estimation of MAOC$_{max}$ was greatest when using QR (τ = 0.90), whereas BL estimates at
5 and 10% intervals were similar to QR (τ = 0.50), and those estimated from linear regression (Table 3). The use of the median
percentile QR highlights the closeness of linear regression predictions being more indicative of mean values, thus
underestimating SOC sequestration potential. The BL estimate of Feng et al. (2013), 0.89 ± 0.05, was nearly double their linear
regression; this was not the case in our study. BL analysis uses a subset of data to estimate, in this case, an upper limit, the
data set used by Feng et al. (2013) had a wider spread of measured MAOC of 0.9 to 71.7 g C kg$^{-1}$ soil, compared to1.72 to



18.29 g C kg⁻¹ soil in our UK soils. Therefore, the upper subset of data was composed of higher values giving a steeper slope and demonstrates that the C sequestration estimate generated by BL analysis is biased by the range of data.

210       The strength of using QR analysis is that it makes no assumptions of homogeneity of variance and uses the entire data set to estimate the upper limit of a response. The measured MAOC in the UK sites lacks homogeneity of variance (Fig. 4), where the variation in the measured MAOC increases with the proportion of fine fraction. Therefore, of the methods explored in this study for our grassland soils, we consider the QR estimates of maximum MAOC to be the most robust.

       Our work and previous studies have estimated maximum MAOC on the basis of mass proportion of fine fraction,
however other parameters such as mineralogy, soil microbial community, environmental conditions (e.g precipitation, Table 2) and land management, can significantly influence MAOC stabilisation (Cotrufo et al., 2015; Kallenbach et al., 2016). In fact, previous studies have included the effect of mineralogy on specific surface area to refine estimates (Beare et al., 2014; Feng et al., 2013; Six et al., 2002). However, this is based on the assumption that SOM coats minerals in a single monolayer but they may also form in multi layers (Kleber et al., 2007). A balance must be found between ease and accuracy of estimate,
in order to utilise sequestration in soils as a $CO_2$ drawdown mechanism.

## 4.2 Composition of SOC in UK grasslands

The C:N ratio of MAOC was 9.84 ± 1.00 (mean ± standard deviation) falling within the typical C:N range of fungi (4.5 to 15), whilst bacteria have a lower C:N ratio of 3 to 5 (Cotrufo et al., 2019), suggesting that the MAOC in the grasslands is
predominantly of fungal origin. Negative correlation between MAOC C:N ratio and mass proportion of fine fraction in a sample (Table 2) suggests there may be a greater contribution of bacterial derived MAOC in soils with a greater proportion of fine fraction.

       Our soils showed statistically significant positive correlations between mass proportion of fine fraction and measured MAOC in all soil types, except for leptosols and luvisols. However, these soil types exhibited the greatest proportion of total
SOC stored as MAOC. Luvisols have a high base saturation facilitating more MAOC stabilisation via complexation of organic ligands by free $Ca^{2+}$ (Chen et al., 2020). Identifying soils where greater proportion of total SOC is stored as MAOC is important in identifying where MAOC needs to be protected, but also where it can be enhanced.

       The MAOC accounted for 4.5 to 50.12% of total SOC in the selected UK sites (Fig. 1). This is lower than values reported in the literature. Cai et al. (2016) found MAOC (< 53 μm) made up 48 to 71% of total SOC in Chinese grasslands,
and in Germany MAOC (< 20 μm) was 49 to 68% of total SOC (Wiesmeier et al., 2014). The lower proportion of MAOC of total SOC found in our UK soils suggests that either the potential for additional C sequestration is high, or C is preferentially stabilised by other means. In grasslands, physical stabilisation may be more dominant due to grassland flora species having fine roots, promoting aggregate formation (O'Brien and Jastrow, 2013; Rasse et al., 2005). Saturation behaviour has been

found to differ between SOC stabilisation mechanisms (Gulde et al., 2008; Stewart et al., 2009), therefore understanding
saturation processes of different soil fractions beyond the fine fraction, may be useful to provide a better estimate of SOC
sequestration potential in UK grasslands.

### 4.3 Effect of sward age on MAOC

It was anticipated that for fields of an older sward age, a greater proportion of total SOC would be MAOC as tillage breaks up
macroaggregates making MAOC available for mineralisation. However, the proportion of total SOC that is MAOC was not
consistently higher in the oldest field, and in some instances was significantly less, such as Aberystwyth (Table A2). When
grouped in five year intervals, significant differences in C:N ratio of MAOC, the proportion of fine fraction in a sample (%)
by mass., and MAOC (g C kg$^{-1}$ soil) were found between age groups (Table 5), however, there was no consistent trend in the
results. This data does not support the hypothesis that older swards will have a greater proportion of SOC stabilised in the fine
soil fraction, and a reduced potential for additional C sequestration. The high density of fine roots contributing to aggregate
formation suggests physical protection is likely a dominant stabilisation process in grasslands, however previous work has
found no effect of sward age or the frequency of grassland reseeding on the % C in differing aggregate fractions (> 2000 μm,
250–2000 μm, 53–250 μm and < 53 μm) (Carolan and Fornara, 2016; Fornara et al., 2020). The impact of reseeding disturbance
may be offset due to the high density of roots in grasslands by facilitating aggregate reformation. Additionally, dissolved
organic carbon (DOC) from below ground inputs is more efficiently stabilised as MAOC than above ground DOC (litter
leachate). The narrow rhizosphere to bulk soil ratio in grasslands, means that this below ground pathway is of greater
importance for both total SOC and MAOC (Sokol and Bradford, 2019). This may make MAOC in grasslands more resilient
to disturbance events.

### 5. Conclusions

Estimating the long-term sequestration of soil C in the fine fraction is difficult due to the lack of reliable methodologies that
can be widely applied to all soils. Our study has demonstrated that the Hassink (1997) linear regression equation is not suitable
to estimate maximum MAOC in a range of UK grassland soils. The significantly different slopes across the UK demonstrate
the variability of the effect of proportion of fine fraction in a sample and current MAOC. Therefore, caution should be applied
to estimates of maximum MAOC obtained using the Hassink (1997) equations in instances where it may not accurately reflect
MAOC stabilisation processes of the soil *in situ*. If estimating maximum MAOC using the proportion of fine fraction and
current MAOC, the use of QR is recommended to overcome shortfalls of linear regression. However, such a simple estimate
is unlikely to accurately reflect the dynamics of MAOC stabilisation, and additional research is required to elucidate parameters

which balance resource inputs and predictive power. Such work would help to accurately assess the feasibility of achieving soil carbon sequestration targets. In temperate soils such as the UK grassland soils studied here, MAOC only made up a small

proportion of total SOC suggesting a dominance of other stabilisation processes. Whilst there was an inconsistent effect of sward age in this study, further research to understand dominant SOC stabilisation and its resilience in response to land management is essential to ensure that current SOC is not only enhanced but also protected.

**Author contribution**

KCP, SB, JMC, RMR and EMB formulated the research question and study design. KCP conducted the experimental work,

data analysis, and prepared the manuscript draft.  All authors contributed to editing and reviewing of the manuscript.

**Data Availability**

All data resulting from this study are available from the authors upon request to Sarah Buckingham (sarah.buckingham@sruc.ac.uk)

**Competing Interest**

The authors declare that they have no conflict of interest.

**Acknowledgements**

We acknowledge funding from Business Environment, Industry and Strategy, Ricardo AEA, Scottish Government's Strategic Research Programme and the Global Academy of Agriculture and Food Security, University of Edinburgh. We are grateful to John Parker and Lydia Guo for their assistance in both the field and laboratory. We also thank Mr Steve Freeman for technical

assistance, and Margaret Glendining and Sarah Perryman for access to information and data from the Electronic Rothamsted Archive (e-RA). The Rothamsted Long-term Experiments National Capability (LTE-NCG) is supported by the UK Biotechnology and Biological Sciences Research Council (BBS/E/C/000J0300) and the Lawes Agricultural Trust. The authors are also grateful to collaborators at the local field sites for supplying soil samples and management information.






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





**Table 1 Summary of UK grassland site characteristics.**

| Site | Age range (years) | Land Use[a] | Mean Annual Temperature (ºC)[b] | Mean Annual Rainfall (mm)[a] | Elevation (m.a.s.l) | WRB Soil Type[c] | Soil Texture[c] |
|---|---|---|---|---|---|---|---|
| **Aberystwyth (52°25'N 04°02'W)** | 2 to 33 | UpG | 9.5 to 11 | 1000 | 20 to 65 | ST, CM | Clay to sandy loam |
| **Crichton (55°02'N 03°35'W)** | 1 to 20 | DP | 9.5 to 9.9 | 1100 | 5 to 50 | CM | Clay loam to sandy loam |
| **Easter Bush (55°51'N 03°52'W)** | 3 to 6 | MG | 6 to 9 | < 700 | 215 to 265 | GL | Clay loam to sandy loam |
| **Harpenden (51°48'N 00°22'W)** | 22 to 179 | UnG | 9.5 to 10.5 | 700 | 120 to 130 | LV | Silty clay loam |
| **Hillsborough (54°27' N 6°04' W)** | 1 to 37 | DP | 8.5 to 10 | 900 | 120 | CM | Clay loam |
| **Kirkton (56°25'N 04°39'W)** | 1 to 35 | UpG | 8 to 9.4 | 2528 | 163 to 170 | PZ | Clay loam to sandy loam |
| **Llangorse (51°55'N 03°16'W)** | 2.5 to 25 | MG | 8 to 10 | 1000 | | CM | Loam/ Clay to Silty loam |
| **Myerscough (53°51'N 02°46'W)** | 2 to 48.4 | MG | 9 to 10.5 | 1000 | 8 to 15 | GL | Clay to sandy loam |
| **Overton (51°48'N 02°08'W)** | 3 to 50 | MGO | 9 to 11 | 800 | 240 to 276 | LP | Clay loam to silty loam |
| **Plumpton (50°54'N 00°04'W)** | 1 to 20 | MG | 9.5 to 11 | 800 | 49 to 85 &160 to 215 | ST | Clay to clay loam, Chalky clay to chalky loam |

[a] Land Use; DP; Dairy pasture, MG; Mixed grazing; MGO; Mixed grazing organic, UpG; Upland grazing, UnG; Ungrazed.

[b] Mean annual temperature and rainfall estimated from Met Office climatic region summaries, averaged over 1981 to 2010.

[c] World Reference Base (WRB) Soil Type: ST; Stagnosols, CM; Cambisols, GL;Gleysol, LV; Luvisols; PZ; Podzol; LP; Leptosol .Soil type and texture determined from GPS locations and UK Soil Observatory Map viewer.






**Table 2. Correlation matrix of Kendal tau (τ) coefficients for bulk and fine fraction (<20 μm) soil properties, sward age and known environmental parameters.**

| | | | | | | Bulk Soil | | | | | Fine Fraction | | |
|---|---|---|---|---|---|---|---|---|---|---|---|---|---|
| | | Temp. | Prec. | Age | %N | %C | C:N | pH | %SC | MAOC | %N | %C | C:N |
| | Temp. | 1 | | | | | | | | | | | |
| | Prec. | -0.05 | 1 | | | | | | | | | | |
| **Bulk soil** | Age | 0.15 | -0.11 | 1 | | | | | | | | | |
| | %N | 0.23*** | -0.07 | 0 | 1 | | | | | | | | |
| | %C | 0.16* | 0.06 | 0.04 | 0.73*** | 1 | | | | | | | |
| | C:N | -0.25*** | -0.05 | 0.02 | -a | -a | 1 | | | | | | |
| | pH | 0.07 | -0.30*** | -0.07 | -0.04 | 0.03 | 0.02 | 1 | | | | | |
| | %SC | 0.26*** | -0.43*** | 0.14* | 0.12 | 0.12 | -0.01 | 0.21*** | 1 | | | | |
| | MAOC | 0.13* | -0.36*** | 0.1 | 0.26*** | 0.27*** | 0.01 | 0.12 | -a | 1 | | | |
| **Fine Fraction** | %N | -0.32*** | 0.28*** | -0.07 | 0.17** | 0.14* | -0.02 | -0.27*** | -0.47*** | -0.15** | 1 | | |
| | %C | -0.33*** | 0.25*** | -0.09 | 0.18** | 0.17** | 0.06 | -0.25*** | -0.47*** | -a | 0.87*** | 1 | |
| | C:N | -0.21*** | -0.08 | -0.16 | 0.11** | 0.21*** | 0.30*** | -0.05 | -0.15* | -0.02 | -a | -a | 1 |

a No correlation calculated as one variable used to calculate the other.

Age; years since last reseeding event, Temp; median value from the mean annual temperature range (°C), Prec.; mean annual

rainfall (mm), %SC; mass proportion of fine fraction in a sample (%), MAOC; measured mineral associated organic carbon (g MAOC kg$^{-1}$ bulk soil).

Level of significance: * $P < 0.05$, ** $P < 0.01$, ***$P < 0.0001$







**Table. 3. Analyses coefficients for the estimation of max MAOC by linear regression (LR), linear regression with forced zero intercept (LR_0), boundary line (BL) and quantile regression (QR). Lettering indicates slopes which were significantly different within a method ($P < 0.05$).**

| Method | | Slope (± 1 SEM) | *P* slope | Intercept (± 1 SEM) | *P* intercept | RMSE | n |
|---|---|---|---|---|---|---|---|
| **LR** | Hassink, (1997) | 0.37[a] | | 4.07 | | | 40 |
| | All UK | 0.32 ± 0.023[b] | *** | 2.86 ± 0.368 | *** | 2.58 | 129 |
| **LR_0** | Hassink, (1997)[a] | 0.45 ± 0.02 | *** | | | 4.97 | 40 |
| | All UK | 0.47 ± 0.017 | *** | | | 3.13 | 129 |
| **BL** | 5% intervals | 0.48 ± 0.058 | *** | | | 5.89 | 19 |
| | 10% intervals | 0.48 ± 0.070 | *** | | | 6.36 | 15 |
| | 15% intervals | 0.56 ± 0.056 | *** | | | 4.77 | 14 |
| **QR** | QR ($\tau = 0.90$) | 0.92 ± 0.071 | *** | | | 7.90 | 129 |
| | QR ($\tau = 0.50$) | 0.49 ± 0.032 | *** | | | 3.15 | 129 |

RMSE, root mean square error.

Level of significance: *** $P < 0.001$

[a] Data extracted from Hassink, (1997) used to generate slope value with forced zero intercept.






**Table 4. Carbon saturation ratios calculated from the estimated MAOC$_{max}$ by linear regression (LR), linear regression with forced zero intercept (LR_0), boundary line (BL) and quantile regression (QR). Values < 1 indicate unsaturated, = 1 at saturation and > 1 are oversaturated samples.**

| Method | | No. of unsaturated samples (n = 129) | Mean ratio | Median |
|---|---|---|---|---|
| **LR** | Hassink, (1997) | 105 | 0.77 | 0.73 |
| | UK | 75 | 0.98 | 0.94 |
| | UK site specific | 71 | 1 | 0.99 |
| **Forced 0 intercept** | | | | |
| **LR_0** | Hassink (1997) | 30 | 1.52 | 1.44 |
| | UK | 34 | 1.47 | 1.39 |
| | UK site specific | 57 | 1.09 | 1.04 |
| **BL** | 5% | 38 | 1.42 | 1.34 |
| | 10% | 36 | 1.43 | 1.35 |
| | 15% | 50 | 1.22 | 1.15 |
| **QR** | 50$^{th}$ | 38 | 1.4 | 1.32 |
| | 90$^{th}$ | 99 | 0.74 | 0.7 |








**Table 5. Effect of sward age grouped at five year intervals on selected soil properties. Values are means ± standard error of the mean, and different letters indicate age groups which are significantly different ($P < 0.05$), by columns.**

| Age | n | C:N | % SC | MAOC (g C kg$^{-1}$ soil) |
|---|---|---|---|---|
| 0 to 5 | 48 | $10.18 \pm 0.15^a$ | $10.00 \pm 1.41^a$ | $5.68 \pm 0.49^a$ |
| 6 to 10 | 18 | $9.79 \pm 0.26^{ab}$ | $14.47 \pm 1.69^{bc}$ | $8.58 \pm 0.59^b$ |
| 11 to 15 | 15 | $9.33 \pm 0.11^b$ | $15.27 \pm 2.98^{abc}$ | $9.17 \pm 1.66^{ab}$ |
| 16 to 20 | 6 | $10.24 \pm 0.37^{ab}$ | $4.85 \pm 0.68^{ab}$ | $4.47 \pm 0.65^a$ |
| 21+ | 42 | $9.59 \pm 0.12^a$ | $14.72 \pm 1.59^c$ | $7.27 \pm 0.65^{ab}$ |

Age; years since last reseeding event, C:N ratio of the fine fraction, %SC; proportion of fine fraction in a sample (%) by mass.




**Figure 1.** Measured total SOC (g C kg⁻¹ soil) (A) total MAOC (g C kg⁻¹ soil) (B), mass proportion of fine fraction (< 20 μm, %) (C) and relative proportion of measured current MAOC of the total SOC content of the bulk soil (D), for each of the grassland sites; Aberyswyth (A), Crichton (C), Easter Bush (E), Hillsborough (H), Harpenden (Ha), Kirkton (K), Llangorse (L), Myerscough (M), Overton (O) and Plumpton (P). Boxes represent the 25th and 75th percentile, with lines showing the median value. Whiskers show the lowest and highest values with outliers indicates as crosses (> 1.5 times the interquartile range). Lettering indicates significant differences between soils (*P* < 0.05).





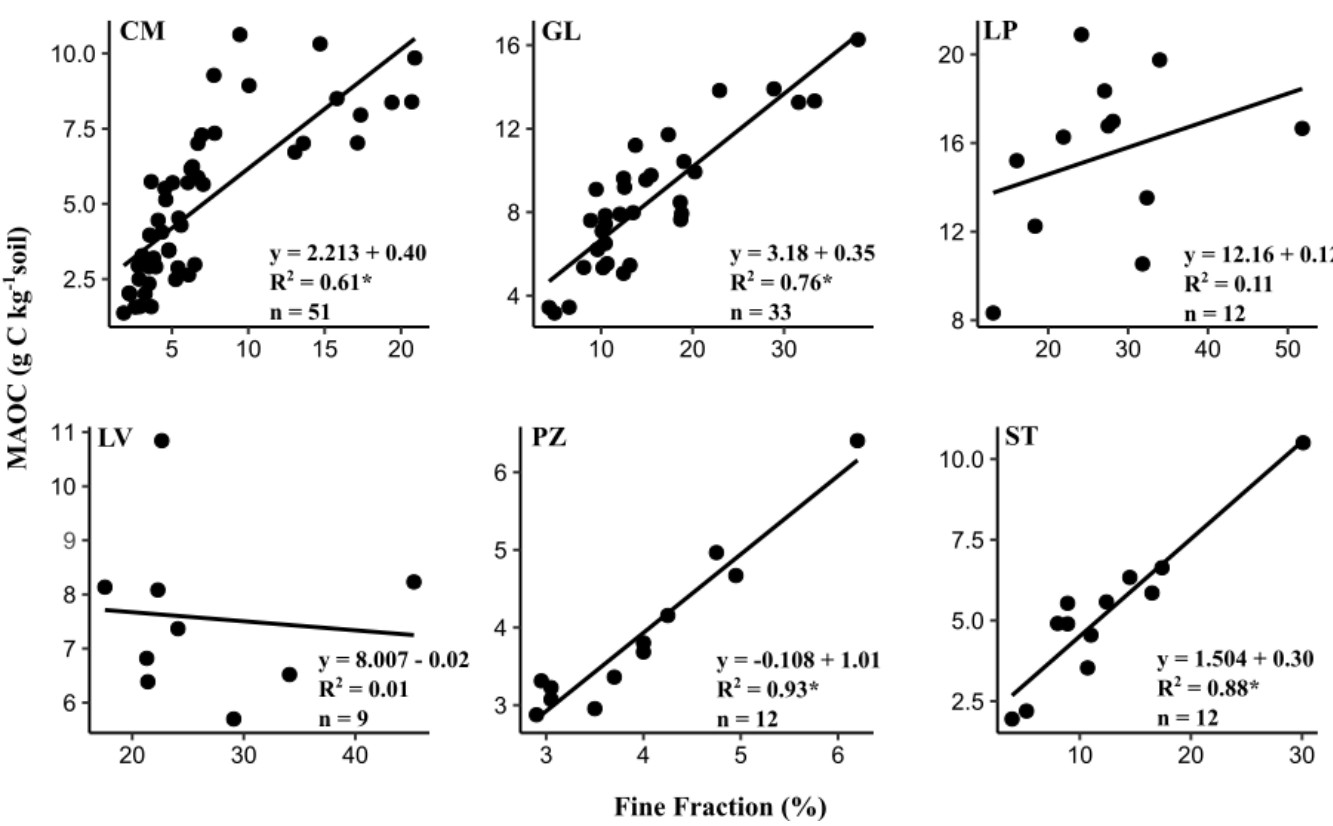

**Figure 2.** Relationships between mass proportion of the fine fraction (%) and MAOC (g C kg$^{-1}$ soil) in the soil types used in this study; cambisols (CM), gleysols (GL), leptosols (LP), luvisols (LV), podzols (PZ) and stagnosols (ST). Asterisks indicate significance at $P < 0.05$.







**Figure 3. Relative proportion of measured current MAOC of the total SOC content of the bulk soil for the different soil types used in this study; cambisols (CM), gleysols (GL), leptosols (LP), luvisols (LV), podzols (PZ) and stagnosols (ST). Lettering indicates significant differences at $P < 0.05$.**





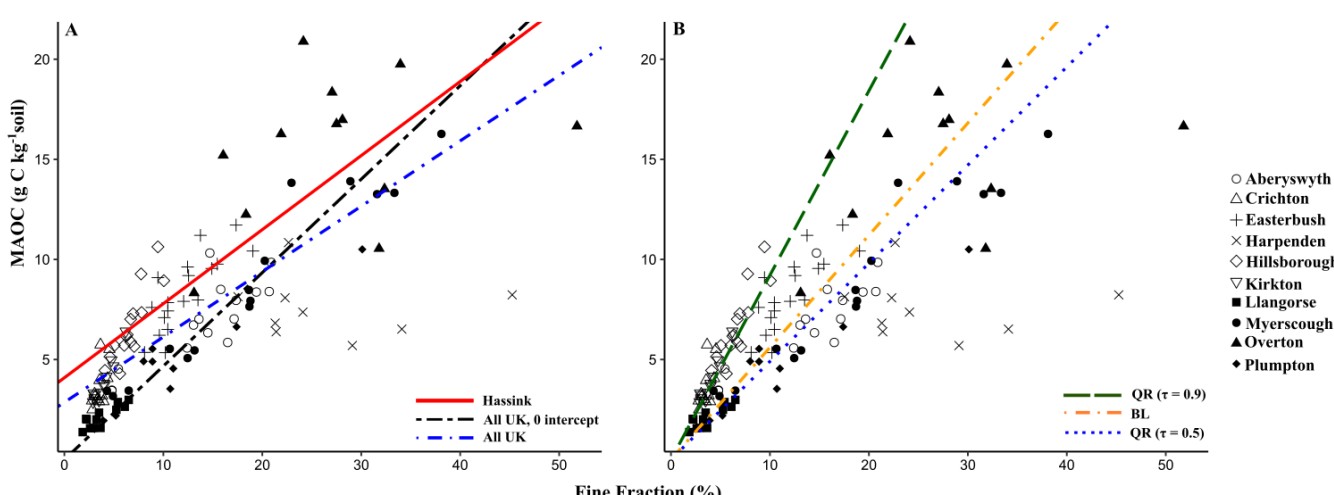

**Figure 4. Measured MAOC (g C kg⁻¹ soil) in relation to mass proportion of fine fraction of a soil sample (%). Line of best fit represent (A) linear regression method of Hassink, (1997) and data from this study, and (B) boundary line (BL) using 15% intervals, and quantile regression analysis (QR) at 90ᵗʰ and 50ᵗʰ percentiles.**






## Appendix A

**Table A1. Bulk soil properties for each UK site. Values are means of ten replicates in each field, ± one standard error of the mean. Except Harpenden where values are means of five replicates per field. Lettering indicates values which are significantly different, within a site ($P < 0.05$).**

| Site | Age (years) | BD[a] | pH | C:N | C (g C kg⁻¹ soil) | C stock (t C ha⁻¹) | N stock (t N ha⁻¹) |
|---|---|---|---|---|---|---|---|
| Aberyswyth | 2 | $1 \pm 0.01^a$ | $5.20 \pm 0.05^a$ | $9.70 \pm 0.05^b$ | $26.95 \pm 0.63^b$ | $73.61 \pm 1.73^b$ | $7.59 \pm 0.16^b$ |
| | 6 | $0.98 \pm 0.04^a$ | $4.70 \pm 0.04^{bc}$ | $9.68 \pm 0.08^b$ | $26.7 \pm 0.82^b$ | $73.86 \pm 2.28^b$ | $7.62 \pm 0.18^b$ |
| | 11 | $0.82 \pm 0.05^b$ | $5.12 \pm 0.06^a$ | $10.46 \pm 0.09^a$ | $29.72 \pm 0.83^b$ | $76.97 \pm 2.15^b$ | $7.36 \pm 0.20^b$ |
| | 31 | $0.74 \pm 0.05^b$ | $4.99 \pm 0.09^{ab}$ | $10.54 \pm 0.22^a$ | $29.4 \pm 1.63^b$ | $74.23 \pm 4.12^b$ | $7.01 \pm 0.31^b$ |
| | 33 | $0.69 \pm 0.03^b$ | $4.18 \pm 0.02^c$ | $10.59 \pm 0.10^a$ | $38.19 \pm 1.97^b$ | $95.67 \pm 4.92^a$ | $9.01 \pm 0.40^a$ |
| Crichton | 1 | $0.92 \pm 0.03$ | $5.14 \pm 0.03^{ab}$ | $12.19 \pm 0.08^{ab}$ | $34.66 \pm 0.66^a$ | $82.40 \pm 1.56$ | $6.76 \pm 0.11^b$ |
| | 3 | $0.99 \pm 0.07$ | $5.65 \pm 0.06^b$ | $11.73 \pm 0.11^{bc}$ | $29.94 \pm 1.37^{ab}$ | $74.63 \pm 3.41$ | $6.36 \pm 0.26^b$ |
| | 15 | $0.93 \pm 0.05$ | $4.77 \pm 0.04^{ac}$ | $9.90 \pm 0.88^c$ | $30.85 \pm 3.06^b$ | $79.73 \pm 7.91$ | $7.98 \pm 0.23^a$ |
| | 20 | $0.93 \pm 0.04$ | $4.54 \pm 0.03^c$ | $13.21 \pm 0.14^a$ | $27.26 \pm 0.87^b$ | $66.62 \pm 2.11$ | $5.04 \pm 0.15^c$ |
| Easter Bush | 3 | $1.02 \pm 0.04^{abc}$ | $5.45 \pm 0.06^{ab}$ | $13.03 \pm 0.11^{bc}$ | $32.46 \pm 1.29^a$ | $93.52 \pm 3.72^a$ | $7.17 \pm 0.26^a$ |
| | 5 | $1.19 \pm 0.03^a$ | $5.44 \pm 0.06^{ab}$ | $12.84 \pm 0.21^{bc}$ | $26.41 \pm 0.54^b$ | $74.45 \pm 1.52^b$ | $5.80 \pm 0.10^{bc}$ |
| | 5 | $0.84 \pm 0.06^c$ | $5.67 \pm 0.04^a$ | $11.74 \pm 0.17^d$ | $27.50 \pm 1.0^b$ | $58.15 \pm 2.12^c$ | $4.94 \pm 0.14^d$ |
| | 6 | $0.96 \pm 0.05^{bc}$ | $5.32 \pm 0.06^b$ | $12.45 \pm 0.13^c$ | $30.46 \pm 1.93^{ab}$ | $71.16 \pm 4.50^{bc}$ | $5.72 \pm 0.36^{cd}$ |
| | 6 | $1.12 \pm 0.05^{ab}$ | $5.81 \pm 0.20^a$ | $14.15 \pm 0.13^a$ | $28.95 \pm 0.99^{ab}$ | $75.69 \pm 2.59^b$ | $5.35 \pm 0.17^{cd}$ |
| | 8 | $1.12 \pm 0.03^{ab}$ | $4.99 \pm 0.04^c$ | $13.43 \pm 0.11^b$ | $33.03 \pm 0.50^a$ | $89.43 \pm 1.34^a$ | $6.66 \pm 0.10^{ab}$ |
| Harpenden | 22 | $1.37 \pm 0.07$ | $7.37 \pm 0.04^a$ | $12.09 \pm 0.2$ | $16.06 \pm 0.59^c$ | $25.37 \pm 0.93^c$ | $3.3 \pm 0.12^c$ |
| | 68 | $1.12 \pm 0.08$ | $5.85 \pm 0.12^{ab}$ | $12.34 \pm 0.08$ | $19.8 \pm 0.63^b$ | $50.49 \pm 1.59^b$ | $4.06 \pm 0.13^b$ |
| | 179 | $1.09 \pm 0.14$ | $5.63 \pm 0.06^b$ | $12.8 \pm 0.26$ | $28.7 \pm 1.47^a$ | $72.98 \pm 3.74^a$ | $5.89 \pm 0.30^a$ |
| Hillsborough | 1 | $1.79 \pm 0.10$ | $6.31 \pm 0.07^a$ | $11.25 \pm 0.12^{ab}$ | $46.68 \pm 2.04$ | $120.16 \pm 5.26^{ab}$ | $10.69 \pm 0.51^{ab}$ |
| | 7 | $1.88 \pm 0.08$ | $5.10 \pm 0.04^b$ | $11.46 \pm 0.11^b$ | $42.85 \pm 1.52$ | $108.86 \pm 3.87^b$ | $9.51 \pm 0.34^{bc}$ |
| | 16 | $1.79 \pm 0.05$ | $5.33 \pm 0.08^b$ | $10.87 \pm 0.06^c$ | $42.36 \pm 1.98$ | $111.63 \pm 5.21^{ab}$ | $10.27 \pm 0.47^{ab}$ |
| | 23 | $1.75 \pm 0.05$ | $4.76 \pm 0.03^c$ | $11.33 \pm 0.09^{ab}$ | $46.44 \pm 1.78$ | $125.43 \pm 4.82^a$ | $11.08 \pm 0.45^a$ |
| | 37 | $1.69 \pm 0.06$ | $5.13 \pm 0.06^b$ | $10.34 \pm 0.77^{ac}$ | $40.90 \pm 3.10$ | $86.04 \pm 6.52^c$ | $8.38 \pm 0.24^c$ |
| Kirkton | 1 | $0.9 \pm 0.04$ | $4.78 \pm 0.04^c$ | $12.13 \pm 0.11^c$ | $27.90 \pm 0.81^c$ | $82.03 \pm 2.39^b$ | $6.77 \pm 0.22$ |
| | 3 | $0.95 \pm 0.04$ | $5.49 \pm 0.06^a$ | $12.61 \pm 0.15^b$ | $36.67 \pm 1.56^a$ | $98.19 \pm 4.17^{ab}$ | $7.79 \pm 0.31$ |





| | | | | | | | |
|---|---|---|---|---|---|---|---|
| | 5 | 0.83 ± 0.06 | 5.15 ± 0.03[b] | 13.56 ± 0.08[a] | 34.83 ± 1.84[ab] | 103.03 ± 5.45[a] | 7.59 ± 0.38 |
| | 35 | 0.97 ± 0.06 | 4.72 ± 0.07[c] | 11.67 ± 0.13[d] | 30.51 ± 1.48[bc] | 90.50 ± 4.38[ab] | 7.72 ± 0.32 |
| Llangorse | 2.5 | 1.01 ± 0.04 | 5.14 ± 0.08[c] | 9.21 ± 0.09 | 17.83 ± 0.42 | 49.75 ± 1.18 | 5.40 ± 0.11[ab] |
| | 5 | 0.93 ± 0.04 | 5.44 ± 0.03[b] | 9.40 ± 0.07 | 18.60 ± 0.45 | 50.80 ± 1.22 | 5.40 ± 0.10[b] |
| | 15 | 0.94 ± 0.06 | 5.68 ± 0.03[a] | 9.36 ± 0.17 | 19.42 ± 0.38 | 53.70 ± 1.06 | 5.74 ± 0.12[ab] |
| | 25 | 1.06 ± 0.03 | 5.54 ±0.07[ab] | 9.16 ± 0.87 | 19.73 ± 2.52 | 55.10 ± 7.05 | 6.18 ± 0.34[a] |
| Myerscough | 2 | 1.22 ± 0.02[ab] | 4.97 ± 0.05[b] | 13.58 ± 0.24[bc] | 27.47 ± 0.65[c] | 82.25 ± 1.96[c] | 6.07 ± 0.15[bc] |
| | 6 | 1.10 ± 0.04[b] | 5.59 ± 0.05[a] | 11.79 ± 0.76[c] | 41.44 ± 2.73[a] | 124.05 ± 8.17[a] | 10.56 ± 0.28[a] |
| | 13 | 0.93 ± 0.05[b] | 5.00 ± 0.20[b] | 13.12 ± 0.43[c] | 44.82 ± 2.34[a] | 134.45 ± 7.01[a] | 10.30 ± 0.71[a] |
| | 34 | 1.29 ± 0.02[a] | 5.99 ± 0.13a | 17.20 ± 1.12[ab] | 37.58 ± 1.45[ab] | 112.46 ± 4.36[ab] | 6.71 ± 0.30[b] |
| | 48.4 | 1.44 ± 0.06[a] | 5.77 ± 0.02[a] | 22.10 ± 1.46[a] | 29.86 ± 1.96[bc] | 88.97 ± 5.85[bc] | 4.03 ± 0.08[c] |
| Overton | 3 | 0.98 ± 0.09[a] | 6.58 ± 0.12[b] | 9.76 ± 0.05[b] | 32.77 ± 0.84[c] | 83.02 ± 2.13[b] | 8.51 ± 0.23[b] |
| | 12 | 0.38 ± 0.03[b] | 6.83 ± 0.03[b] | 10.18 ± 0.12[ab] | 70.18 ± 1.92[a] | 81.20 ± 2.23[b] | 7.99 ± 0.23[b] |
| | 22 | 0.71 ± 0.07[ab] | 7.36 ± 0.04[a] | 10.68 ± 0.39[a] | 59.88 ± 3.86[b] | 132.75 ± 8.56[a] | 12.33 ± 0.39[a] |
| | 50 | 1.74 ± 0.9[a] | 4.63 ± 0.08[c] | 10.14 ± 0.14[ab] | 51.18 ± 2.84[b] | 153.08 ± 8.50[a] | 15.08 ± 0.80[a] |
| Plumpton | 1 | 0.99 ± 0.02[a] | 6.34 ± 0.08[b] | 10.85 ± 0.08ab | 40.92 ± 1.21[b] | 122.21 ± 3.61[b] | 11.26 ± 0.28[b] |
| | 5 | 1.08 ± 0.03a | 7.15 ± 0.06a | 11.27 ± 0.41a | 45.55 ± 0.61[b] | 132.09 ± 1.77[b] | 11.87 ± 0.48[b] |
| | 20 | 0.72 ± 0.04b | 5.38 ± 0.21c | 10.54 ± 0.17b | 58.08 ± 2.36[a] | 163.23 ± 6.62[a] | 15.47 ± 0.56[a] |

[a]Bulk density (BD), means and SEM of six samples, except Harpenden with two samples per field, corrected for stones.





**Table A2. Fine fraction (<20 μm) soil properties for each UK site. Values are means of three replicates in each field, ± one standard error of the mean. Lettering indicates values which are significantly different, within a site ($P < 0.05$).**

| Location | Age (years) | %N | %C | C:N | %SC | MAOC (g C kg⁻¹ bulk soil)[a] | MAOC (% of $SOC_{total}$) |
|---|---|---|---|---|---|---|---|
| Aberyswyth | 2 | 0.48 ± 0.01[b] | 4.16 ± 0.08 | 8.62 ± 0.08[ab] | 19.08 ± 1.04[a] | 7.93 ± 0.45[a] | 0.30 = 0.01[a] |
| | 6 | 0.52 ± 0.04[b] | 4.14 ± 0.30 | 8.05 ± 0.17[b] | 14.47 ± 1.18[ab] | 5.92 ± 0.22[ab] | 0.21 ± 0.02[ab] |
| | 11 | 0.55 ± 0.03[b] | 4.89 ± 0.25 | 8.86 ± 0.06[ab] | 18.02 ± 1.51[ab] | 8.77 ± 0.56[a] | 0.29 ± 0.02[ab] |
| | 31 | 0.61 ± 0.05[ab] | 5.78 ± 0.62 | 9.51 ± 0.30[ab] | 13.78 ± 0.49[b] | 8.02 ± 1.15[a] | 0.27 ± 0.02[ab] |
| | 33 | 0.76 ± 0.03[a] | 7.57 ± 0.37 | 9.96 ± 0.23[a] | 5.02 ± 0.22[c] | 3.81 ± 0.36[b] | 0.10 ± 0.01[b] |
| Crichton | 1 | 1.01 ± 0.06 | 10.53 ± 0.83 | 10.40 ± 0.23[ab] | 4.00 ± 0.45 | 4.24 ± 0.69 | 0.12 ± 0.02 |
| | 3 | 1.15 ± 0.27 | 11.17 ± 2.28 | 9.84 ± 0.35[b] | 3.28 ± 0.23 | 3.75 ± 1.00 | 0.13 ± 0.04 |
| | 15 | 1.02 ± 0.12 | 9.76 ± 1.20 | 9.52 ± 0.12[b] | 3.52 ± 0.26 | 3.38 ± 0.31 | 0.10 ± 0.02 |
| | 20 | 0.82 ± 0.05 | 9.07 ± 0.72 | 11.03 ± 0.24[a] | 3.37 ± 0.3 | 3.01 ± 0.09 | 0.11 ± 0.01 |
| Easter Bush | 3 | 0.65 ± 0.04 | 7.15 ± 0.50 | 11.00 ± 0.13[ab] | 14.38 ± 1.56[ab] | 10.27 ± 1.19[a] | 0.30 ± 0.01 |
| | 5 | 0.65 ± 0.04 | 6.91 ± 0.50 | 10.57 ± 0.06[bc] | 12.17 ± 0.9[ab] | 8.34 ± 0.43[ab] | 0.32 ± 0.02 |
| | 5 | 0.67 ± 0.02 | 6.62 ± 0.23 | 9.83 ± 0.07[c] | 9.55 ± 0.73[b] | 6.32 ± 0.51[b] | 0.23 ± 0.03 |
| | 6 | 0.68 ± 0.03 | 7.81 ± 0.43 | 9.85 ± 0.24[c] | 9.75 ± 0.23[b] | 6.88 ± 1.13[ab] | 0.23 ± 0.01 |
| | 6 | 0.72 ± 0.12 | 7.11 ± 1.31 | 11.43 ± 0.16[a] | 10.58 ± 1.04[b] | 8.22 ± 0.70[ab] | 0.27 ± 0.02 |
| | 8 | 0.59 ± 0.04 | 6.07 ± 0.30 | 10.35 ± 0.28[bc] | 16.47 ± 1.3[a] | 9.91 ± 0.26[ab] | 0.30 ± 0.01 |
| Harpenden | 22 | 0.23 ± 0.01[b] | 1.90 ± 0.04[c] | 8.20 ± 0.26[b] | 36.15 ± 4.77[a] | 6.82 ± 0.75[ab] | 0.42 ± 0.04 |
| | 68 | 0.32 ± 0.01[b] | 3.08 ± 0.06[b] | 9.54 ± 0.21[a] | 22.27 ± 0.92[b] | 6.86 ± 0.28[ab] | 0.36 ± 0.02 |
| | 179 | 0.46 ± 0.03[a] | 4.35 ± 0.36[a] | 9.54 ± 0.12[a] | 20.83 ± 1.64[b] | 9.02 ± 0.91[a] | 0.32 ± 0.01 |
| Hillsborough | 1 | 0.90 ± 0.08 | 8.97 ± 1.14 | 9.91 ± 0.34 | 7.37 ± 0.12 | 6.86 ± 1.93 | 0.14 ± 0.03 |
| | 7 | 1.04 ± 0.06 | 10.23 ± 0.91 | 9.80 ± 0.31 | 8.05 ± 0.08 | 8.15 ± 0.96 | 0.17 ± 0.02 |
| | 16 | 0.99 ± 0.04 | 9.36 ± 0.32 | 9.46 ±0.03 | 6.33 ± 0.19 | 5.92 ± 0.13 | 0.15 ± 0.01 |
| | 23 | 1.15 ± 0.01 | 11.11 ± 0.13 | 9.70 ± 0.18 | 4.58 ± 0.27 | 5.10 ± 0.36 | 0.12 ± 0.01 |
| | 37 | 1.04 ± 0.04 | 10.12 ± 0.35 | 9.76 ± 0.04 | 7.15 ± 0.33 | 7.22 ± 0.10 | 0.27 ± 0.11 |
| Kirkton | 1 | 0.91 ± 0.03 | 9.27 ± 0.12[b] | 10.15 ± 0.24[b] | 3.90 ± 0.1 | 3.62 ± 0.13 | 0.14 ± 0.01 |
| | 3 | 1.01 ± 0.04 | 10.63 ± 0.33[a] | 10.56 ± 0.27[ab] | 3.02 ± 0.03 | 3.20 ± 0.07 | 0.08 ± 0.00 |
| | 5 | 0.88 ± 0.03 | 10.23 ± 0.16[ab] | 11.66 ± 0.31[a] | 4.62 ± 0.95 | 4.75 ± 1.03 | 0.13 ± 0.02 |





|  | 35 | 0.96 ± 0.03 | 9.22 ± 0.40[b] | 9.64 ± 0.39[b] | 4.23 ± 0.42 | 3.93 ± 0.51 | 0.14 ± 0.00 |
|---|---|---|---|---|---|---|---|
| Llangorse | 2.5 | 0.51 ± 0.03[b] | 4.76 ± 0.29[b] | 9.36 ± 0.07 | 6.00 ± 0.32[a] | 2.83 ± 0.10[a] | 0.16 ± 0.01[a] |
|  | 5 | 0.88 ± 0.08[a] | 8.29 ± 0.80[a] | 9.43 ± 0.07 | 2.65 ± 0.43[b] | 2.13 ± 0.11[ab] | 0.11 ± 0.01[ab] |
|  | 15 | 0.67 ± 0.10[ab] | 6.06 ± 0.78[ab] | 9.11 ± 0.24 | 3.23 ± 1.03[b] | 1.81 ± 0.34[b] | 0.09 ± 0.02[b] |
|  | 25 | 0.62 ± 0.06[ab] | 5.32 ± 0.54[b] | 8.60 ± 0.05 | 3.27 ± 0.22[b] | 1.72 ± 0.14[b] | 0.07 ± 0.01[b] |
| Myerscough | 2 | 0.63 ± 0.08 | 6.60 ± 0.76 | 10.43 ± 0.12 | 5.23 ± 0.66[c] | 3.35 ± 0.09[c] | 0.12 ± 0.00[b] |
|  | 6 | 0.49 ± 0.03 | 4.57 ± 0.29 | 9.31 ± 0.23 | 27.50 ± 3.85[a] | 12.39 ± 1.24[a] | 0.31 ± 0.04[a] |
|  | 13 | 0.50 ± 0.05 | 4.83 ± 0.60 | 9.52 ± 0.32 | 30.88 ± 4.39[a] | 14.45 ± 0.92[a] | 0.30 ± 0.02[a] |
|  | 34 | 0.47 ± 0.01 | 4.28 ± 0.13 | 9.12 ± 0.07 | 18.72 ± 0.04[ab] | 8.02 ± 0.24[b] | 0.21 ± 0.02[ab] |
|  | 48.4 | 0.47 ± 0.02 | 4.47 ± 0.36 | 9.58 ± 0.36 | 12.08 ± 0.74[bc] | 5.35 ± 0.14[bc] | 0.17 ± 0.02[b] |
| Overton | 3 | 0.42 ± 0.03[c] | 3.57 ± 0.31[c] | 8.45 ± 0.18[b] | 38.65 ± 6.58 | 13.57 ± 1.77 | 0.41 ± 0.05 |
|  | 12 | 0.88 ± 0.05[a] | 8.52 ± 0.59[a] | 9.64 ± 0.11[a] | 20.70 ± 2.41 | 17.45 ± 1.75 | 0.25 ± 0.02 |
|  | 22 | 0.61 ± 0.02[b] | 6.36 ± 0.18[b] | 10.36 ± 0.15[a] | 19.85 ± 4.39 | 12.52 ± 2.50 | 0.23 ± 0.07 |
|  | 50 | 0.63 ± 0.05[b] | 6.23 ± 0.29[b] | 10.04 ± 0.45[a] | 29.50 ± 2.23 | 18.29 ± 0.86 | 0.34 ± 0.04 |
| Plumpton | 1 | 0.35 ± 0.02[b] | 3.81 ± 0.18[b] | 10.87 ± 0.09 | 19.50 ± 5.61 | 7.23 ± 1.74 | 0.18 ± 0.05 |
|  | 5 | 0.36 ± 0.04[b] | 4.19 ± 0.49[b] | 11.76 ± 0.91 | 6.60 ± 2.08 | 2.56 ± 0.49 | 0.06 ± 0.01 |
|  | 20 | 0.56 ± 0.02[a] | 5.96 ± 0.23[a] | 10.75 ± 0.62 | 8.60 ± 0.3 | 5.11 ± 0.21 | 0.08 ± 0.01 |

%SC; mass proportion of fine fraction in a sample (%).

[a] MAOC (g C kg$^{-1}$ bulk soil) accounts for the proportion of fine fraction per kilogram of bulk soil.







**Table A3. Linear regression coefficients for the estimation of maximum MAOC (g C kg $^{-1}$ soil). Lettering indicates slopes which are significantly different ($P < 0.05$).**

| Site | Slope (± 1 SEM) | *P* slope | Intercept (± 1 SEM) | *P* intercept | RMSE | n |
|---|---|---|---|---|---|---|
| **Aberyswyth** | 0.33 ± 0.059[bc] | *** | 2.28 ± 0.892 | * | 1.11 | 15 |
| **Crichton** | 1.14 ± 0.470[abcd] | * | -0.44 ± 1.684 | Ns | 0.79 | 12 |
| **Easter Bush** | 0.49 ± 0.094[d] | *** | 2.33 ± 1.172 | Ns | 1.10 | 18 |
| **Harpenden** | -0.02 ± 0.07[a] | Ns | 8.01 ± 1.837 | ** | 1.42 | 9 |
| **Hillsborough** | 0.97 ± 0.148[d] | *** | 0.16 ± 1.02 | Ns | 0.84 | 15 |
| **Kirkton** | 1.01 ± 0.088[abcd] | *** | -0.11 ± 0.357 | Ns | 0.26 | 12 |
| **Llangorse** | 0.29 ± 0.055[abc] | *** | 1.03 ± 0.225 | *** | 0.27 | 12 |
| **Mysercough** | 0.40 ± 0.031[bcd] | *** | 1.07 ± 0.669 | Ns | 1.14 | 15 |
| **Overton** | 0.12 ± 0.109[cd] | Ns | 12.16 ± 3.142 | ** | 3.35 | 12 |
| **Plumpton** | 0.30 ± 0.042[ab] | *** | 1.45± 0.573 | * | 0.82 | 9 |

RMSE: Root mean square error.

Level of significance: * $P < 0.05$, ** $P < 0.01$, ***$P < 0.001$







**Table A4. Linear regression coefficients for the estimation of maximum MAOC (g C kg$^{-1}$ soil) with a forced zero intercept. Lettering indicates slopes, that are significantly different ($P < 0.05$).**

| Site | Slope (± 1 SEM) | *P* | RMSE | n |
|------|-----------------|-----|------|---|
| **Aberyswyth** | $0.47 \pm 0.024^{bc}$ | *** | 1.357 | 15 |
| **Crichton** | $1.02 \pm 0.067^{cdef}$ | *** | 0.796 | 12 |
| **Easter Bush** | $0.67 \pm 0.024^{e}$ | *** | 1.231 | 18 |
| **Harpenden** | $0.26 \pm 0.035^{a}$ | *** | 2.739 | 9 |
| **Hillsborough** | $0.99 \pm 0.033^{f}$ | *** | 0.842 | 15 |
| **Kirkton** | $0.98 \pm 0.0197^{def}$ | *** | 0.265 | 12 |
| **Llangorse** | $0.52 \pm 0.035^{abcdef}$ | *** | 0.474 | 12 |
| **Mysercough** | $0.45 \pm 0.016^{b}$ | *** | 1.255 | 15 |
| **Overton** | $0.52 \pm 0.055^{bcd}$ | *** | 5.297 | 12 |
| **Plumpton** | $0.39 \pm 0.030^{ab}$ | *** | 1.141 | 9 |

RMSE: Root mean square error.

Level of significance: * $P < 0.05$, ** $P < 0.01$, ***$P < 0.001$