# Peer review of "Estimating maximum fine fraction organic carbon in UK grasslands"

_Biogeosciences, 2020_

## Referee Comment (RC1) · Emanuele Lugato (Referee) · 1 Sep 2020

The manuscript under review investigates the mineral associated organic carbon (MAOC) distribution in grassland soils of varying sward age, across the UK. The authors compared the Hassink's reference equation to calculate the saturation capacity against alternative methods, which showed a more accurate assessments of carbon sequestration potential. The paper is of good quality, with a robust methodology and well written and developed in each sections. I don't have any major concerns but, rather, some points of discussion as following:

The forced intercept to 0 is generally suggested to avoid the paradox of having MAOM without any fine (silt and clay) fractions. However, in my experience with very large

datasets, I have never seen a soil without any fine fraction (at least temperate soils covered with any type of vegetation). It seems that the saturation equation is a type of function where the x domain is always >0. Indeed, the authors forced the intercept to 0 using the BL and QR methods, therefore, it would be worth to have a more in depth elaboration of this choice.

In the paragraph in line 255, the authors reported: "The C:N ratio of MAOC was 9.84 $\pm$ 1.00 (mean $\pm$ standard deviation) falling within the typical C:N range of fungi (4.5 to 15) whilst bacteria have a lower C:N ratio of 3 to 5 (Cotrufo et al., 2019), suggesting that the MAOC in the grasslands is predominantly of fungal origin." Indeed, this is an erroneously interpretation as MAOM is not entirely composed of living microbial biomass. C:N around 9 is on the average of European grassland (Figure 3 of Cotrufo et al., 2019), while other systems 'fungal-dominated' such a coniferous forests have a much higher C:N ratio. By the way, it would be interesting to know if C:N of MAOM differs significant across sites.

The difference between the Hassink's and UK equation implicitly suggests that a universal saturation equation likely does not exist, but many equations are controlled by interacting factors as mineralogy, soil microbial community etc. This is concept is developed around line 215 but the conclusion of the paragraph is quite elusive. I would encourage the authors to developed 'a way forward paragraph' that can guide a future research. I imagine, for instance, incubation experiments with unsaturated soils (according to those equations) where excess of high quality inputs are applied to see their 'real' saturation level. In this context, I wonder if authors can produce a plot of MAOM vs estimated C input, which may reveal (or not) some interesting correlations.

In the conclusion, the QR method is recommended but is not indicated at which quantile level. This makes a substantial difference in the relative proportion of saturated soils (table 4) and, hence, a possible perception of policy priorities. Are the soils mostly saturated or not? Is the index robust enough to provide management guidance? As is it, the conclusion left me a little bit hanging.

Line 78, hypothesis ii: I see also the way around. Since MAOC is less sensitive to disturbance (than POM), the ratio MAOC/SOC is negatively related to sward age. In other words, long-aged sward grasslands accumulate more POC , lowering the ratio MAOC:SOC. The table 5 reports only the absolute values.

Line 115: Is not clear if the comparison of MAOC across sites treats the 'site as random factor (One Random Factor ANOVA).

Line 237-238. This statement does not explain the lower MAOC proportion in UK grassland compared to other 'grassland' sites. Was the MAOC separation method the same?

Table 3: please, add the r2 for completeness

Figure 2: please, add x (independent variable) in the equations

---

## Referee Comment (RC2) · Steffen A. Schweizer (Referee) · 16 Sep 2020

The authors present an interesting dataset and thoughts. However, it is not so clear what the take home messages are: The influence of soil types on <20 $\mu$m OC? One-time physical disturbance might not play a major role for <20 $\mu$m OC? The evaluation of other unifactorial empirical linear models? In its current form the manuscript contains a slightly confusing mixture. Based on the interesting data I suggest to do a major shift and set a new focus beyond unifactorial models. In Table 2 there are so many highly significant correlations between various factors. These should be explored further and related with carbon stocks to develop a higher potential of the manuscript and advance insights into potential processes that govern soil organic carbon stocks in UK grasslands.
Major issues I could identify:

- As described in line 98, the so-called mineral-associated OC 'MAOC' is isolated by sieving. The authors should provide evidence of mineral association or reword this.

- As authors write themselves in line 200 the soil properties vary beyond only the proportion of fine fraction, this should be explored further as stated above. In Fig. 4 it seems that the relationship exists only across sites but not within sites.

- The use of abbreviations sometimes makes the reading hard; this should be minimized as much as possible (soil type names for example)

- As described in line 247, the high variability of the proportion of the fine fraction might seriously confound the analysis of sward age. Is it possible to correct for this using a multifactorial model?

Further issues:

Line 15: Introduction very general, should be more specific towards the research questions

Line 19: What are specific challenges with this?

Line 46: What is the specific soil management involved?

Line 48: Other forms of OC would be probably also mineralized, why especially 'MAOC'?

Line 51: Please add that such estimation of a 'protective capacity' is empirical

Line 143: One decimal would probably be sufficient

Line 187: "very little further testing [. . .] in other soils" There are many papers cited in the manuscript that do exactly this to my understanding

Line 194: Such validity tests probably would include grassland soils which might not always be available (as paired site), also given that there are many other specific factors

[Figure]

– what do you specifically propose?

Line 214: Interesting paragraph but far apart from the data presented here, what can you conclude and contribute to the discussion based on the data in this manuscript?

Line 223: Add reference of original measurements

Line 225: I disagree with such simplified relation. Fungal origin should be verified with another biomarker approach. High C:N could result from root input and particulate OM (as briefly mentioned in line 238). Could C:N results be influenced by mineral N fertilizer?

Line 237: What could this "other means" be?

Line 255: Could you add a literature reference here?

Line 445, Figure 1: The data is repeated in the Appendix. Should be present only at one spot.

Line 455, Figure 2: To improve the comparison between panels, I suggest to put similar x and y scales. Also add the significance level and remove regressions from the Figure when not significant.

---

## Author Comment (AC1) · 14 Oct 2020

**Response to Reviewer 1 Comments on Manuscript, bg-2020-273**

**Estimating maximum mineral associated organic carbon in UK grasslands**

Kirsty C. Paterson, Joanna M. Cloy, Robert. M. Rees, Elizabeth M. Baggs, Hugh Martineau, Dario Fornara, Andrew J. Macdonald, and Sarah Buckingham

We thank the reviewers for their comments and evaluation of our manuscript. Please find below our response to comments made by reviewer 1. Reviewer's comments are in black text, and our responses are in blue text, changes to the manuscript are highlighted in yellow. Line numbers refer to the revised manuscript (marked version) below.

**Overall Comments:**

The manuscript under review investigates the mineral associated organic carbon (MAOC) distribution in grassland soils of varying sward age, across the UK. The authors compared the Hassink's reference equation to calculate the saturation capacity against alternative methods, which showed a more accurate assessments of carbon sequestration potential. The paper is of good quality, with a robust methodology and well written and developed in each sections. I do not have any major concerns but, rather, some points of discussion as following:

The forced intercept to 0 is generally suggested to avoid the paradox of having MAOM without any fine (silt and clay) fractions. However, in my experience with very large datasets, I have never seen a soil without any fine fraction (at least temperate soils covered with any type of vegetation). It seems that the saturation equation is a type of function where the x domain is always >0. Indeed, the authors forced the intercept to 0 using the BL and QR methods, therefore, it would be worth to have a more in depth elaboration of this choice.

We agree with this comment, which was outlined in the introduction, lines 73 to 74: "Using a forced zero intercept overcomes the contradiction of a positive intercept indicating the presence of MAOC without any fine soil fraction (Beare et al., 2014; Feng et al., 2013; Liang et al., 2009)".

The following has been added to section 2.3.1, line 168.

"Forcing the intercept to zero overcomes the paradox of having C stabilised as MAOC without any fine fraction in the soil."

In the paragraph in line 255, the authors reported: "The C:N ratio of MAOC was $9.84 \pm 1.00$ (mean $\pm$ standard deviation) falling within the typical C:N range of fungi (4.5 to 15) whilst bacteria have a lower C:N ratio of 3 to 5 (Cotrufo et al., 2019), suggesting that the MAOC in the grasslands is predominantly of fungal origin." Indeed, this is an erroneously interpretation as MAOM is not entirely composed of living microbial biomass. C:N around 9 is on the average of European grassland (Figure 3 of Cotrufo et al., 2019), while other systems 'fungal-dominated' such a coniferous forests have a much higher C:N ratio. By the way, it would be interesting to know if C:N of MAOM differs significant across sites.

The C:N of MAOM does differ significantly between the selected sites. An extra figure, E has been added to the panel in figure 1, see below. The following has been added to the results section 3.1, lines 190 to 193.

"Soil C:N ratio was positively correlated with fine fraction C:N (0.30, $P < 0.0001$), Table 2, however there was no relationship between bulk soil C:N ratio and proportion of fine fraction (data not shown). The fine fraction C:N ratio was significantly different between the sites, Figure 1, however the mean of all the data showed little deviation, $9.84 \pm 1.00$ (mean $\pm$ standard deviation)."

As suggested by both reviewers the comment regarding the C:N ratio and potential origin of the OC has been removed.

[Figure]

**Figure 1. Measured total SOC (g C kg⁻¹soil) (A) total fine fraction organic carbon (g C kg⁻¹soil) (B), mass proportion of fine fraction (< 20 μm, %) (C), relative proportion of measured fine fraction organic carbon of the total SOC content of the bulk soil (D) and fine fraction C:N ratio (E), for each of the grassland sites; Aberyswyth (A), Crichton (C), Easter Bush (E), Hillsborough (H), Harpenden (Ha), Kirkton (K), Llangorse (L), Myerscough (M), Overton (O) and Plumpton (P). Boxes represent the 25th and 75th percentile, with lines**

**showing the median value. Whiskers show the lowest and highest values with outliers indicates as crosses (> 1.5 times the interquartile range). Lettering indicates significant differences between soils ($P < 0.05$).**

60

The difference between the Hassink's and UK equation implicitly suggests that a universal saturation equation likely does not exist, but many equations are controlled by interacting factors as mineralogy, soil microbial community etc. This is concept is developed around line 215 but the conclusion of the paragraph is quite elusive. I would encourage the authors to developed 'a way forward paragraph' that can guide a future research. I imagine,

65  for instance, incubation experiments with unsaturated soils (according to those equations) where excess of high quality inputs are applied to see their 'real' saturation level. In this context, I wonder if authors can produce a plot of MAOM vs estimated C input, which may reveal (or not) some interesting correlations.

The following has been added to the discussion, lines 301 to 307.

70  "Whilst we consider the quantile regression at the 90th percentile method to provide the most robust estimate of maximum fine fraction OC in the sites studied, further experimental work to test the saturation level of these soils, would help to validate this. Incubation studies which force an unsaturated soil to its 'saturation' level and the effect of influencing variables, mentioned above will help to elucidate the factors controlling fine fraction OC saturation. In particular further empirical evidence of how to manipulate fine fraction OC stabilisation processes in a way that

75  is practical for grassland management to promote the formation of new organo-mineral associations, and understanding their stability will be important for establishing the true potential of additional carbon sequestration across managed grasslands."

A plot as advised has been produced, however it was felt to be more beneficial to present it on a site basis. The y

80  axis is not consistent in each graph as it was impossible to tell the data points apart in some instances using one scale for all. See lines 270 to 282 in the revised manuscript.

"When examining the estimated OC input versus existing fine fraction OC using estimates generated by quantile regression at the 90[th] percentile a positive correlation between current fine fraction OC and estimated C input

85  (Kendall tau ($\tau$);0.323, P < 0.001), was observed for the entire data set. However, this was not the case at the site

level, see Fig. A1. Where in some instances increasing fine fraction OC (g C kg$^{-1}$ soil) was associated with increased estimated C input until saturation, such as Aberyswyth, Myerscough and Plumpton. Therefore, despite a higher fine fraction OC contents these samples are furthest from saturation. In contrast the opposite was true for Crichton and Hillsborough (and Harpenden, Kirkton and Overton, although not statistically significant) implying that for these sites samples with a higher fine fraction OC are closer to saturation. It is unclear why this is the case particularly as in all sites, bar Harpenden, there is a positive regression between mass proportion of the fine fraction and fine fraction OC (Table A3). Meaning that higher fine fraction OC is also associated with higher mass proportion of the fine soil fraction. It is likely that the OM input to the soils with the higher mass proportion of fine fraction is insufficient to bridge the gap between current and estimated maximum fine fraction OC, as it is not possible to identify any other effect due to pedogenic or environmental conditions measured in this work."

[Figure]

**Figure A1. Estimated fine fraction OC input (g C kg$^{-1}$ soil) compared to measured fine fraction OC (g C kg$^{-1}$ soil) in each of the sites studied. The estimated OC input was predicted using quantile regression at the 90$^{th}$ percentile.**

105 In the conclusion, the QR method is recommended but is not indicated at which quantile level. This makes a substantial difference in the relative proportion of saturated soils (table 4) and, hence, a possible perception of policy priorities. Are the soils mostly saturated or not? Is the index robust enough to provide management guidance? As is it, the conclusion left me a little bit hanging.

The QR percentile has been specified throughout the discussion where use of QR is recommended.

110

Lines 268 and 269 have been amended and incorporated into the following paragraph, see changes highlighted in yellow below, to make it clear that of the methods explored in this work, the QR at the 90th percentile is the most robust. But all of the methods over simplify the dynamics of MAOC accrual, therefore a greater understanding of how, and if, MAOC stabilisation processes occur through further research will make it possible to consider if this

115 is a priority for carbon sequestration policies.

"Therefore, of the methods explored in this study for our grassland soils, we consider the quantile regression at the 90th percentile estimate of maximum fine fraction OC to be the most robust. This method results in the greatest number of unsaturated samples (Table 4) suggesting great potential for additional sequestration."

120

This position is then summarised in the conclusion, lines 350 to 356.

"After exploring various univariate estimation methods we recommend the use of quantile regression at the 90th percentile to overcome the shortfalls of linear regression. However, such a simple estimate is unlikely to accurately reflect the dynamics of fine fraction OC stabilisation. This work has helped to identify some key parameters which

125 play a role in fine fraction OC stabilisation, such median annual temperature, mean annual precipitation, bulk soil %C and %N and fine fraction %N. Further work to understand how these parameters influence fine fraction OC dynamics, will help to accurately assess the feasibility of achieving soil carbon sequestration targets".

**Specific Comments:**

130 Line 78, hypothesis ii: I see also the way around. Since MAOC is less sensitive to disturbance (than POM), the ratio MAOC/SOC is negatively related to sward age. In other words, long-aged sward grasslands accumulate more POC , lowering the ratio MAOC:SOC. The table 5 reports only the absolute values.

This would be the alternate hypothesis to the one we present in hypothesis ii. Section 4.3, lines 310 to 330 have been updated, highlighted in yellow below, to include discussion of this alternative hypothesis, and the potential caveats due to small sample size of the 16 to 20 years age group.

"It was anticipated that for fields of an older sward age, a greater proportion of total SOC would be stabilised as fine fraction OC, as tillage breaks up macroaggregates making OC in the fine fraction available for mineralisation. Alternatively, fine fraction OC is less sensitive to disturbance than particulate organic matter (POM), resulting in the accumulation of POM as the fine fraction OC pool remains stable, if sufficiently saturated. The results seem to support neither hypothesis. The proportion of total SOC stabilised in the fine fraction was not consistently higher in the oldest field, and in some instances was significantly less, such as Aberystwyth (Table A2). When grouped in five year intervals, significant differences in C:N ratio of the fine fraction, the proportion of fine fraction in a sample (%) by mass, measured fine fraction OC (g C kg$^{-1}$ soil) and the relative proportion of measured fine fraction OC of the total SOC content of the bulk soil were found between age groups (Table 5), however, there was no consistent trend in the results. This data does not support the hypothesis that older swards will have a greater proportion of SOC stabilised in the fine soil fraction, and a reduced potential for additional C sequestration. Equally there was no negative correlation between sward age and the proportion of total SOC stabilised which would be supportive of the alternate hypothesis. From the data it appears that fine fraction OC makes up a greater proportion of SOC with increasing sward age when comparing the less than 5 years, 6 to 10 and 11 to 15 years age groups. However there is a significant decrease in the amount of SOC that is stabilised in the fine fraction in the 16 to 20 years group, this is likely due to fields in this age range originating from Crichton, Hillsborough and Plumpton, which have some of the lowest mass proportion of the fine fraction (Figure 1, C). The sward age analysis may also be confounded by the variation of the proportion of fine fraction, particularly on soil properties influenced by mass proportion of fine fraction such as %C and %N and current fine fraction OC (g C kg$^{-1}$ soil). However, it was not possible to conduct robust ANCOVA's with a grouping variable with more than two levels. It may be possible to elucidate the relationship better from a wider study with more samples per age group as our 16 to 20 years group only has 9 values compared to 48 in the less than 5 years group."

160

Line 115: Is not clear if the comparison of MAOC across sites treats the 'site as random factor (One Random Factor ANOVA).

For the comparison of fine fraction OC across the sites (results displayed Figure 1, panel B) Kruskal test and post hoc Dunn testing was used as the data lacks homogenous variance and therefore does not meet the requirements of

165    an ANOVA. This is outlined in section 2.3.

Line 237-238. This statement does not explain the lower MAOC proportion in UK grassland compared to other 'grassland' sites. Was the MAOC separation method the same?

170    Under restructuring to clarify the focus of the manuscript as suggested by reviewer 2, this statement has been removed.

Table 3: please, add the r2 for completeness

Done.

175

Figure 2: please, add x (independent variable) in the equations

Done.

[revised manuscript text omitted]

---

## Author Comment (AC2) · 14 Oct 2020

**Response to Reviewer 2 Comments on Manuscript, bg-2020-273**

**Estimating maximum mineral associated organic carbon in UK grasslands**

Kirsty C. Paterson, Joanna M. Cloy, Robert. M. Rees, Elizabeth M. Baggs, Hugh Martineau, Dario Fornara, Andrew J. Macdonald, and Sarah Buckingham

We thank the reviewers for their comments and evaluation of our manuscript. Please find below our response to comments made by reviewer 2. Reviewer's comments are in black text, and our responses are in blue text, changes to the manuscript are highlighted in yellow. Line numbers refer to the revised manuscript (marked version) below.

**Overall Comments:**

The authors present an interesting dataset and thoughts. However, it is not so clear what the take home messages are: The influence of soil types on <20 m OC? One time physical disturbance might not play a major role for <20 m OC? The evaluation of other unifactorial empirical linear models? In its current form the manuscript contains a slightly confusing mixture. Based on the interesting data I suggest to do a major shift and set a new focus beyond unifactorial models. In Table 2 there are so many highly significant correlations between various factors. These should be explored further and related with carbon stocks to develop a higher potential of the manuscript and advance insights into potential processes that govern soil organic carbon stocks in UK grasslands.

We thank the reviewer for the comments and have made revisions to the manuscript in order to clarify the take home messages. The introduction has been restructured to provide sufficient background information of the aims and objectives. The concluding paragraph has been amended to focus on the key take away points with respect to the hypotheses outlined in the introduction. We agree that correlations between the variables presented are interesting, and the suggested exploration would provide further insights into carbon stocks in UK grasslands.

The recommendation as described has been carried out as part of a sister project that focusses on soil C stocks. A manuscript based on that work is currently being prepared. Therefore, the aim of this separate study was to take a subset of samples to examine the suitability of the Hassink, (1997) linear regression equation, and to explore methods of estimating maximum fine fraction OC, and the impact of reseeding events on current fine fraction OC and predicted maximums. Taking the reviewer's advice on board, we believe that the new structure better frames the aims and objectives and take home messages.

**Specific Comments:**

- As described in line 98, the so-called mineral-associated OC 'MAOC' is isolated by sieving. The authors should provide evidence of mineral association or reword this.

Whilst the mineral associated organic matter fraction, and C within, i.e. MAOC, has been defined to refer to just OM adsorbed to minerals, or include OM encapsulated in micro-aggregates (Kögel-Knabner et al., 2008), resulting in size based definitions ranging from 0 to 20 µm or 0 to 53 µm (Lavallee et al., 2020; Six et al., 2002). The term "OC in the fine fraction (< 20 um)" has been used throughout as it was not possible to provide empirical evidence of mineral association, and also to overcome assumptions of fraction size which may be associated with the term MAOC. The title has also been amended to reflect this change to "Estimating maximum fine fraction organic carbon in UK grasslands"

- As authors write themselves in line 200 the soil properties vary beyond only the proportion of fine fraction, this should be explored further as stated above. In Fig. 4 it seems that the relationship exists only across sites but not within sites.

The following has been added to line 251 to 253.

"The range of reported values, and differences across the UK sites (Tables A3 and A4), suggests that the effect of the proportion of fine fraction of a sample on fine fraction OC is not consistent and likely reflects differences in

50    pedogenic and environmental conditions, and land management. It may be that the use of the mass proportion of fine fraction to predict maximum fine fraction OC is only suited on larger scales, rather than smaller, site specific scales, as indicated by the variability in this study."

- The use of abbreviations sometimes makes the reading hard; this should be minimized as much as possible (soil
55    type names for example).

This has been addressed throughout the text. Where abbreviations are used in tables or figures they have been clearly defined in the captions.

- As described in line 247, the high variability of the proportion of the fine fraction might seriously confound the
60    analysis of sward age. Is it possible to correct for this using a multifactorial model?

The reviewer makes an interesting suggestion and the proportion of fine fraction was indeed variable within and among the different sward age groups. We tried to investigate this further by means of ANCOVA testing, however as the data does not satisfy the assumptions the use of a robust ANCOVA. The package available in R (WRS2) to do this only allows for two levels within the grouping variable (Mair and Wilcox, 2020). In order to
65    acknowledge potential confounding effect, on variables linked to mass proportion of the fine fraction, the following has been added to the discussion, lines 326 to 329.

"The sward age analysis may also be confounded by the variation of the proportion of fine fraction, particularly on soil properties influenced by mass proportion of fine fraction such as %C and %N and current fine fraction
70    OC (g C kg$^{-1}$ soil). However, it was not possible to conduct robust ANCOVA's with a grouping variable with more than two levels."

75   Line 15: Introduction very general, should be more specific towards the research questions

The introduction has been edited to provide more focused background to the objectives of the study, with a brief background to carbon sequestration, methods of predicting maximum fine fraction OC, (objectives i and ii), and the effects of grassland re-seeding events, (objective iii). See updated manuscript.

80   Line 19: What are specific challenges with this?

The following has been added to line 66 to elaborate this point in the introduction.

"There is a need for validity checks to determine the suitability of the Hassink, (1997) linear regression equation to predict maximum fine fraction OC of the soils in the respective studies. Without this sequestration potentials may be both over and underestimated."

85

The following has been added to the discussion to clarify the issues, lines 231 to 234.

"However, the Hassink (1997) linear regression equation, equation has been used to estimate sequestration potentials, without prior testing to determine its applicability to the soils in question (e.g. Angers et al., 2011;

90   Chen et al., 2019; Lilly and Baggaley, 2013; Wiesmeier et al., 2014). This may have potentially over or underestimated sequestration potential, which may have repercussions for decisions made regarding land management"

Line 46: What is the specific soil management involved?

95   Details have been added to line 70.

"The high levels of disturbance associated with re-seeding events by mould board ploughing and harrowing in particular, result in changes in soil structure, notably the breaking up of aggregates, nutrient cycling and SOC mineralisation (Carolan and Fornara, 2016; Drewer et al., 2017; Soussana et al., 2004)."

100 Line 48: Other forms of OC would be probably also mineralized, why especially 'MAOC'?

We agree that other forms of OC will also become more accessible for mineralisation due to the disturbance associated with reseeding events. However, as the focus of this work is on the mineral associated pool, we highlight that the disruption of aggregates in particular makes MAOC more accessible for microbial mineralisation. The following amendments have been made to better emphasis this and how the disturbance of

105 other forms of OC may cause MAOC mineralisation, lines 75 to 80.

"Organo-mineral associations form the basis of microaggregates (Baldock and Skjemstad, 2000), and thus the destruction of aggregates makes the organo-mineral stabilised OC in the fine fraction, more accessible for mineralisation by the soil microbial community. Additionally, the release of other organic carbon pools may

110 induce a priming effect, potentially enhancing the losses from the typically stable mineral associated OC in the fine fraction. The long-term effect of such re-seeding event on SOC dynamics is understudied, it is therefore important to understand how disturbance might affect OC in the fine fraction, and thus the SOC sequestration ability of managed grasslands."

115 Line 51: Please add that such estimation of a 'protective capacity' is empirical

Done.

Line 143: One decimal would probably be sufficient

Done.

120

Line 187: "very little further testing [: : :] in other soils" There are many papers cited in the manuscript that do exactly this to my understanding

The meaning here is in reference to studies which used the linear equation (e.g. Angers et al., 2011; Chen et al., 2019; Lilly and Baggaley, 2013; Wiesmeier et al., 2014) without assessing its applicability to the specific soils in question. It is not intended to refer to studies which have tried to improve the method of estimating maximum fine fraction OC. The following changes have been made, lines 231 to 234.

"However, the Hassink (1997) linear regression equation has been used to estimate sequestration potentials, without prior testing to determine its applicability to the soils in question (e.g. Angers et al., 2011; Chen et al., 2019; Lilly and Baggaley, 2013; Wiesmeier et al., 2014). This may have potentially over or underestimated sequestration potential, which may have repercussions for decisions made regarding land management."

Line 194: Such validity tests probably would include grassland soils which might not always be available (as paired site), also given that there are many other specific factors – what do you specifically propose?

Given the results of the study, if someone wished to try to estimate maximum fine fraction OC using a univariate regression method, we suggest they develop their own linear regression equation by determining OC content of the fine fraction and mass proportion of fine fraction of a subset of the sample. Rather than assuming that the Hassink, (1997) is valid for their soils, see line 243.

Line 214: Interesting paragraph but far apart from the data presented here, what can you conclude and contribute to the discussion based on the data in this manuscript?

The paragraph has been amended to provide a better discussion focused on the results from this study. Lines 283 to 294.

Line 223: Add reference of original measurements

Done.

Line 225: I disagree with such simplified relation. Fungal origin should be verified with another biomarker approach. High C:N could result from root input and particulate OM (as briefly mentioned in line 238). Could C:N results be influenced by mineral N fertilizer?

Under the restructuring of the manuscript this comment no longer applies.

Line 237: What could this "other means" be?

Lines 331 to 333, have been amended to the following;

"Fine fraction OC only accounted for 4.5 to 50.12% indicating high OC storage in other soil pools such as POM, or different aggregate fractions. The fine roots of grassland flora species promote aggregate formation (O'Brien and Jastrow, 2013; Rasse et al., 2005), which may be a dominant stabilisation process in grasslands."

Line 255: Could you add a literature reference here?

Done.

Line 445, Figure 1: The data is repeated in the Appendix. Should be present only at one spot.

The data in the appendix gives readers the opportunity to examine the results between the fields within each site, which is not possible from figure 1, so, respectfully, we suggest to retain both.

Line 455, Figure 2: To improve the comparison between panels, I suggest to put similar x and y scales. Also add the significance level and remove regressions from the Figure when not significant.

The x and y-axis have been adjusted and significance levels added. However, regression details have been retain for transparency.

[revised manuscript text omitted]

---

## Author Response (AR2)

**Response to Reviewers 1 and 2 on Manuscript, bg-2020-273**

**Estimating maximum fine fraction organic carbon in UK grasslands**

Kirsty C. Paterson, Joanna M. Cloy, Robert. M. Rees, Elizabeth M. Baggs, Hugh Martineau, Dario Fornara,
5    Andrew J. Macdonald, and Sarah Buckingham

We thank the reviewers for their comments and evaluation of our manuscript. Please find below our response to comments made by reviewers 1 and 2. Reviewer's comments are in black text, and our responses are in blue text,
10   changes to the manuscript are highlighted in yellow. Line numbers refer to the revised manuscript (marked version) below.

**Reviewer 1**

The authors further improved the paper considering the suggestion/comments raised, therefore I think that can be
15   accepted for publication. I have only some minor suggestions and an issue not completely taken from the last review.

One of the take-home message of the paper is about the unsuitability of Hassink's equation in UK conditions but, then, the recommended Quantile Regression (90th percentile) gives similar absolute and mean ratio of unsaturated
20   sites (Table 4). Is there any contradiction in the message?

The absolute number of unsaturated samples are similar, 105 and 99 for Hassink's equation in the UK and the Quantile Regression (90[th] percentile) equation, generated using the data from this study. However, as shown in figure 4, the samples deemed oversaturated, above the solid red line in panel A, and above the green dashed line in
25   panel B, differ. For instance, the samples from Overton with a fine fraction % between 20 and 30 are deemed oversaturated using the Hassink equation, whilst they are undersaturated when the maximum fine fraction organic carbon is estimated by quantile regression. We therefore do not believe this to be contradictory as the samples that make up the absolute numbers presented in table 4 differ.

30   Moreover, I still think there is nothing wrong to fit saturation with the intercept (as done by Hassink, Six and others), since the saturation equation is likely a type of function where the x domain is always >0. At least, the recommended QR90 may be fitted also with the intercept; maybe it does not change too much when selecting the 90th percentile (as indeed regressing all the data). Unless the Authors considered the intercept approach totally wrong, this aspect can be mentioned.

We agree that with large data sets it is unlikely that a sample will have no fine fraction, and therefore x is always > 0. However, in this instance we have opted to follow the logic that it is not possible to detect organic carbon in the fine fraction, when there is none, and chose to use a forced zero intercept.

40

Line 33: likely 'protection' in aggregates is more appropriate than stabilization

Changed

Line 77: The sentence about priming is very generic without any reference.

45   The sentence has been changed to the following, lines 72 and 73.

"Organo-mineral associations form the basis of microaggregates (Baldock and Skjemstad, 2000), and thus the destruction of aggregates makes the organic carbon protected within the aggregates more accessible for mineralisation by the soil microbial community. This may result in the increased mineralisation of existing SOC,
50   known as the priming effect (Kuzyakov et al., 2000)"

Line 191 and Fig.1. As C:N of fine fraction was inserted, I would suggest to add also C:N of bulk soil since there is room in the figure layout.

55   The C:N of the bulk soil has been added to the figure 1, panel F and the caption has been updated. Line 161 has been changed to the following.

"The fine fraction and bulk soil C:N ratios were significantly different between the sites, Figure 1. However, the mean value of the fine fraction showed little deviation, $9.84 \pm 0.94$ (mean $\pm$ standard deviation). Full details of all the measured properties of bulk and fine fraction, per field are presented in Table A1."

60

[Figure]

**Figure 1. Measured total SOC (g C kg⁻¹soil) (A) total fine fraction organic carbon  (g C kg⁻¹soil) (B), mass proportion of fine fraction (< 20 μm, %) (C),  relative proportion of measured fine fraction organic carbon of the total SOC content of the bulk soil (D), fine fraction C:N ratio (E) and bulk soil C:N ratio (F), for each of the grassland sites; Aberyswyth (A), Crichton (C), Easter Bush (E), Hillsborough (H), Harpenden (Ha), Kirkton (K), Llangorse (L), Myerscough (M), Overton (O) and Plumpton (P). Boxes represent the 25ᵗʰ and 75ᵗʰ percentile, with lines showing the median value. Whiskers show the lowest and highest values with outliers indicates as crosses (> 1.5 times the interquartile range). Lettering indicates significant differences between soils ($P < 0.05$).**

Line 270: at least in the appendix (figure caption), a small explanation of how C inputs are calculated is needed if authors want to comment this figure. By the way, is not clear what the unit "estimated fine fraction OC input OC (g kg-1 C kg-1)" is.

The calculation method has been added to the caption of figure 1A.

**"Figure A1. Estimated fine fraction OC input (g C kg⁻¹ soil) compared to measured fine fraction OC (g C kg⁻¹ soil) in each of the sites studied. The estimated fine fraction OC input (g C kg⁻¹ soil) was calculated by subtracting the maximum fine fraction OC (g C kg⁻¹ soil) from the current fine fraction OC (g C kg⁻¹ soil). The maximum fine fraction OC (g C kg⁻¹ soil) was estimated using the quantile regression equation ($\tau = 0.90$), where, maximum fine fraction OC = 0.92 multiplied by the mass proportion fine fraction (%)."**

Lines 351-2: isn'it the quantile regression applied also linear?

Yes, this is correct, the words "least squares" have been added to correct the sentence.

 **Reviewer 2**

The authors have improved many aspects of the manuscript, providing further details, and strengthening the structure and conclusions on this compelling topic. I have some further comments, which I offer as suggestions, towards a foreseeable acceptance of this work for publication.

95 It would be nice to see a stronger case made for the quantile and boundary line expression based on its much higher empirical estimates of a maximum fine fraction OC. If only the higher data points are used, in what property do they differ from the lower data points? Is there a correlation of the soil properties with the differences from the maximum fine fraction OC? As the authors emphasize the regression method has "repercussions for decisions made regarding land management". If only the higher data points are considered, how can this be translated into practice? 100 Why should the maximum fine fraction OC be ranked even higher considering that its variability also increases? Can you identify specific soil types or properties that can help to adapt and improve local decisions on land management?

In the conclusions we summarise our support for the use of quantile regression to estimate maximum fine fraction 105 OC. The reasons for this are outlined in the discussion, primarily due to its statistical robustness.

We have explored the data to try and identify soil properties which could help explain why a sample is further or closer to saturation, as suggested by examining those at either end. However, it was not possible to elucidate any properties which may be causing this. This was discussed in lines 229 to 240, from the perspective of additional 110 OC input to reach saturation.

In terms of land management, we felt that the sites were too diverse to give some practical recommendations. In order to do so it would be good to examine paired grasslands, in terms of soil and environmental characteristics, but with contrasting management, for example organic versus conventional to tease out what management 115 processes can be altered to maximise fine fraction OC. The following was added to lines 254 to 257.

"Further work investigating grasslands with similar soil types and textures, and environmental conditions, but contrasting management in terms of fertiliser regimes, grazing densities, sward composition and management, may help to elucidate management factors which can be used to increase fine fraction OC and explain the differences observed in this work."

Given the high variability of the maximum fine fraction OC found in larger datasets and how it responds to a multitude of soil properties and management factors, the applicability of a linear regression estimate may be debated but still provide a valid approach depending on the individual research questions. From my perspective, the authors should reword the statements in l. 66 and 231 towards more specific and constructive details when referring to previous studies that used the Hassink linear regression equation. Can you quantify the mentioned over or underestimation as well as the variability of the data in the 90th percentile?
Overall, nice work.

The sentences have been changed to the following, in order to convey the aim of this work 
[revised manuscript text omitted]

---

## Author Response (AR3)

**Response to associate editor on Manuscript, bg-2020-273**

**Estimating maximum fine fraction organic carbon in UK grasslands**

Kirsty C. Paterson, Joanna M. Cloy, Robert. M. Rees, Elizabeth M. Baggs, Hugh Martineau, Dario Fornara, Andrew J. Macdonald, and Sarah Buckingham

10 Dear authors,

Thank you for your revised manuscript, which I am pleased to accept for publication in Biogeosciences.

I indicated the option for technical corrections because I noticed that the revised text could still be improved. For example the
15 sentence on l. 58-60 in the revised manuscript:

This style of assessment... : do you mean This (type of) approach?

OC in UK soils, without this sequestration...: I suggest to start a new sentence "Without this,...".

...sequestration potentials may be both over and underestimated: ...carbon sequestration potential may be either over- or underestimated.

Please take this opportunity to carefully check the manuscript before uploading your final files.

Kind regards,

Sara

We thank the associated editor for their recommendation of technical correction prior to publication. Please see the marked manuscript below, where some changes have been made.

[revised manuscript text omitted]